# Iterative Foundation Model Fine-Tuning on Multiple Rewards

**Pouya M. Ghari**
Biogen

**Simone Sciabola**
Biogen

**Ye Wang**[*]
Biogen

## Abstract

Fine-tuning foundation models has emerged as a powerful approach for generating objects with specific desired properties. Reinforcement learning (RL) provides an effective framework for this purpose, enabling models to generate outputs that maximize a given reward function. However, in many applications such as text generation and drug discovery, it can be suboptimal to optimize using a single reward signal, as multiple evaluation criteria are often necessary. This paper proposes a novel reinforcement learning-based method for fine-tuning foundation models using multiple reward signals. By employing an iterative fine-tuning strategy across these rewards, our approach generalizes state-of-the-art RL-based methods. We further provide a theoretical analysis that offers insights into the performance of multi-reward RL fine-tuning. Experimental results across diverse domains including text, biological sequence, and small molecule generation, demonstrate the effectiveness of the proposed algorithm compared to state-of-the-art baselines.

## 1 Introduction

Foundation models have emerged as powerful tools capable of performing a wide range of tasks. Trained on large-scale datasets, they acquire broad knowledge that enables their application across diverse domains. To better align a foundation model with the specific preferences of a downstream task, fine-tuning can be applied to improve both performance and task alignment. Given access to a reward model or a preference dataset, reinforcement learning offers an effective framework for fine-tuning foundation models and large language models (LLMs) to better align with downstream tasks [46, 41, 2]. Preference criteria used to evaluate the quality of responses generated by LLMs can vary, and in some cases, it may not be possible to derive a single reward or preference. Furthermore, these criteria can sometimes conflict with one another, making it difficult to summarize them into a single, unified preference metric. For example, human preferences can be diverse and conflicting with one another, such as the trade-off between harmlessness and helpfulness [3]. As another example, LLMs can be used to generate novel small molecules for drug design [50, 22, 35]. In such applications, candidate molecules are often evaluated based on multiple criteria [24, 44]. In such cases, fine-tuning foundation models on multiple objectives becomes essential.

Multi-objective reinforcement learning can be employed to address diverse rewards and preferences. Existing methods in the literature primarily follow two approaches. The first approach combines all reward signals corresponding to different objectives into a single scalar reward [31], which is then used to fine-tune the foundation model. The second approach involves fine-tuning the foundation model separately and *independently* for each objective to obtain a set of expert policy networks, each specialized for a specific objective. These expert policies are then merged to form a unified policy [42], effectively acting as an ensemble with the aim of capturing knowledge from all experts. However, combining reward signals into a single objective may prevent the model from learning objective-specific skills. This can result in high performance variance across objectives, particularly

---

[*]Corresponding Author: ye.wang@biogen.com

39th Conference on Neural Information Processing Systems (NeurIPS 2025).

when a minority of objectives conflict with a majority that are more similar. On the other hand, merging expert policies into a single policy may lead to suboptimal performance across some or all objectives, especially when there is significant divergence among the expert policies due to conflicting objectives.

This paper introduces a novel multi-objective reinforcement learning method for fine-tuning foundation models. To enable the model to acquire objective-specific skills, the proposed algorithm fine-tunes the foundation model separately for each objective, resulting in an expert policy network for each one. However, this fine-tuning is *not* performed independently. To control variance among the expert policies, the algorithm breaks the fine-tuning process into smaller steps and performs it *iteratively*. After each step, the expert policies are merged into a single policy, which is then used as the starting point for the next round of objective-specific fine-tuning. We show that the proposed method can be interpreted as a generalization of both reward-combining and expert-policy-merging approaches. Furthermore, we analyze the convergence properties of the algorithm, providing theoretical insights into its performance. The contributions of this paper are summarized as follows:

- We propose a novel and generalized algorithm that offers greater flexibility than reward-combining and expert-merging baselines, leading to improved performance.
- We provide a theoretical analysis of the proposed algorithm, offering insights into its properties.
- We conduct experiments across diverse tasks including small molecule design, DNA sequence generation, and text summarization, to demonstrate the effectiveness of the proposed method.

## 2    Related Works

**RLHF.** Reinforcement learning with human feedback (RLHF) has been extensively studied in the literature and has demonstrated promising results across various applications [30, 51, 10, 21]. In the context of aligning foundation models with human preferences, RLHF emerges as a compelling approach, as it enables the model to interact with humans feedback to their preferences [26]. Several approaches have been proposed to improve the performance and efficiency of RLHF [13]. The safety of RLHF has been studied by [12]. The alignment of multimodal large language models with human preferences has been investigated by [56, 60]. However, these works typically assume that preferences can be captured using a single feedback signal. In practice, preferences can be diverse, and relying on a single signal may be insufficient to represent this variability.

**Multi-Objective Reinforcement Learning.** The problem of multi-objective optimization has attracted significant attention in reinforcement learning [20, 24]. Several studies have extended deep reinforcement learning techniques to address multi-objective problems [53, 1, 34]. However, focusing on a single mode of the reward function can limit the ability of multi-objective reinforcement learning methods to learn objective-specific skills and may reduce the diversity of the generated outputs. Moreover, when fine-tuning large foundation models, the scalability of multi-objective reinforcement learning becomes critical, potentially making traditional approaches unsuitable for such large-scale applications. To fine-tune foundation models on multiple objectives, the Rewarded Soups [42] method has been proposed. It follows an expert-merging approach, where a separate model is fine-tuned for each objective and then linearly combined to obtain a unified policy. To improve the performance of expert-merging methods particularly in molecular design applications, a more complex merging algorithm was introduced in [7].

**Supervised Fine-Tuning.** Multi-dimensional attributes can be used as conditioning signals for supervised fine-tuning of LLMs [14, 43]. This strategy has been applied to the problem of fine-tuning LLMs on multiple objectives in [52]. By appending the rewards associated with the objectives of interest to the prompts, supervised fine-tuning approach in [52] enables the LLM to learn the relationships between prompt–response pairs and the corresponding multi-objective reward space.

## 3    Preliminaries

This section defines the problem of fine-tuning language models with multiple objectives and reviews relevant approaches.

## 3.1 Multi-Objective Fine-Tuning Problem

Let $\pi_{\boldsymbol{\theta}}$ denote the policy of a language model parameterized by $\boldsymbol{\theta}$, and let $\pi_{\text{ref}}$ represent the initial (reference) policy of the model. Given a prompt $\boldsymbol{x}$, the policy $\pi_{\boldsymbol{\theta}}$ generates a response $\boldsymbol{y}$ by sampling from the distribution $\pi_{\boldsymbol{\theta}}(\boldsymbol{y} \mid \boldsymbol{x})$. Suppose there are $N$ objectives, $R_1, \ldots, R_N$, where for each objective $R_i$, the goal is to learn a policy $\pi_{\boldsymbol{\theta}}$ that minimizes the corresponding loss function $\mathcal{L}_i(\pi_{\boldsymbol{\theta}})$, defined as:

$$\mathcal{L}_i(\pi_{\boldsymbol{\theta}}) = -\mathbb{E}_{\boldsymbol{x} \sim p_{\text{data}}, \, \boldsymbol{y} \sim \pi_{\boldsymbol{\theta}}}[R_i(\boldsymbol{x}, \boldsymbol{y}, \pi_{\boldsymbol{\theta}}, \pi_{\text{ref}})]. \tag{1}$$

Each $R_i$ can represent an objective commonly used in reinforcement learning-based methods such as PPO or DPO. For example, in the context of Reinforcement Learning from Human Feedback (RLHF), assuming access to a reward model $r_\phi$ parameterized by $\phi$, the objective $R_i$ may be defined as:

$$R_i(\boldsymbol{x}, \boldsymbol{y}, \pi_{\boldsymbol{\theta}}, \pi_{\text{ref}}) = r_\phi(\boldsymbol{x}, \boldsymbol{y}) - \beta \log \frac{\pi_{\boldsymbol{\theta}}(\boldsymbol{y}|\boldsymbol{x})}{\pi_{\text{ref}}(\boldsymbol{y}|\boldsymbol{x})}, \tag{2}$$

where $\beta \geq 0$ is a regularization coefficient that penalizes deviation from the reference policy, ensuring that the learned policy does not diverge excessively from $\pi_{\text{ref}}$. Assume that the weight $0 < w_i < 1$ represents the preference for objective $R_i$, where the weights satisfy $\sum_{i=1}^{N} w_i = 1$. In this work, we assume that the preference weights $w_i$ are known for each objective $R_i$. Under this assumption, the problem of multi-objective model fine-tuning can be formulated as:

$$\boldsymbol{\theta}^* = \arg\min_{\boldsymbol{\theta}} \sum_{i=1}^{N} w_i \mathcal{L}_i(\pi_{\boldsymbol{\theta}}). \tag{3}$$

This optimization problem can be addressed using stochastic gradient descent (SGD) techniques. In the remainder of this section, we review two approaches for solving it.

## 3.2 Reward Combining

One approach to solving the optimization problem in equation 3 is to apply reinforcement learning with combined rewards. We refer to this approach as MORLHF in this paper. Let the policy $\pi_{\boldsymbol{\theta}}$ be optimized over $T$ steps, with $\boldsymbol{\theta}_t$ denoting the policy parameters at step $1 \leq t \leq T$. At each step $t$, MORLHF defines a combined single-objective loss as:

$$\mathcal{L}_{\text{MORLHF}}(\pi_{\boldsymbol{\theta}_t}) = -\mathbb{E}_{\boldsymbol{x} \sim p_{\text{data}}, \, \boldsymbol{y} \sim \pi_{\boldsymbol{\theta}_t}} \left[ \sum_{i=1}^{N} w_i R_i(\boldsymbol{x}, \boldsymbol{y}, \pi_{\boldsymbol{\theta}_t}, \pi_{\text{ref}}) \right]. \tag{4}$$

Using the loss function in equation 4, the parameters are updated via gradient descent as:

$$\boldsymbol{\theta}_{t+1} = \boldsymbol{\theta}_t - \eta \nabla_{\boldsymbol{\theta}} \mathcal{L}_{\text{MORLHF}}(\pi_{\boldsymbol{\theta}_t}), \tag{5}$$

where $\eta$ is the learning rate. It is worth noting that multi-objective reinforcement learning can be implemented in various ways through reward combination, with the formulation in equation 4 being just one of them.

## 3.3 Rewarded Soups

An alternative approach to solving the problem in equation 3 is the Rewarded Soups method. This technique optimizes the policy $\pi_{\boldsymbol{\theta}}$ over $T$ steps with respect to each objective $R_i$, yielding a set of parameters $\boldsymbol{\theta}_i$. Specifically, at each step $t$, the parameters are updated as follows:

$$\boldsymbol{\theta}_{i,t+1} = \boldsymbol{\theta}_{i,t} - \eta \nabla_{\boldsymbol{\theta}} \mathcal{L}_i(\pi_{\boldsymbol{\theta}_{i,t}}). \tag{6}$$

After $T$ steps, the parameter $\boldsymbol{\theta}_i = \boldsymbol{\theta}_{i,T}$ is obtained. The final policy $\pi_{\boldsymbol{\theta}_{\text{RS}}}$ is formed by merging the set of expert policies $\{\pi_{\boldsymbol{\theta}_i}\}_{i=1}^{N}$. Each $\pi_{\boldsymbol{\theta}_i}$ is treated as an expert trained on objective $R_i$, and the merged policy acts as an ensemble of these experts. A common merging strategy is to take a weighted linear combination of the parameters:

$$\boldsymbol{\theta}_{\text{RS}} = \sum_{i=1}^{N} \lambda_i \boldsymbol{\theta}_i \tag{7}$$

where each $0 \leq \lambda_i \leq 1$ is a weight associated with the $i$-th objective, satisfying $\sum_{i=1}^{N} \lambda_i = 1$. These weights $\lambda_i$ can be optimized to minimize the loss in equation 3. One approach is to randomly sample candidate weight sets using Monte Carlo methods and select the one yielding the lowest loss. However, this can be computationally expensive. A simpler and more efficient alternative is to set $\lambda_i = w_i$, thereby weighting each expert policy in proportion to its corresponding objective preference.

Comparing MORLHF (Subsection 3.2) with Rewarded Soups (Subsection 3.3), we observe key differences in their approaches. MORLHF optimizes a combined reward signal, aiming to directly learn a policy that balances multiple objectives. In contrast, Rewarded Soups trains separate expert policies for each objective and then constructs the final policy by merging these experts. Because MORLHF does not explicitly specialize in any individual objective, the resulting policy may exhibit high performance variance across different objectives. Conversely, while Rewarded Soups ensures that each expert is well-optimized for its corresponding objective, significant variance among the experts themselves can lead to a merged policy that performs poorly across all objectives.

# 4 Proposed Iterative Fine-Tuning with Multiple Objectives

As discussed in Section 3, MORLHF may exhibit high performance variance across objectives, while Rewarded Soups may experience significant variance among expert policies. This section introduces the proposed approach for fine-tuning models on multiple objectives. By iteratively training expert policies for individual objectives and merging them, the proposed method offers a principled way to mitigate both performance variance across objectives and variances among expert policies.

## 4.1 Algorithm

The proposed algorithm learns an expert policy corresponding to each reward. Let $\boldsymbol{\theta}_{i,t}$ denote the parameters of the policy associated with the $i$-th objective at optimization step $t$. Every $m$ steps where $m$ is an integer hyperparameter the expert policy parameters $\boldsymbol{\theta}_{i,t}$ are merged to produce an updated shared parameter vector $\boldsymbol{\rho}_t$. This merged parameter is then assigned to all expert policies, synchronizing them before continuing individual optimization. To reduce computational complexity, a subset of $M \leq N$ objectives can be selected uniformly at random at each merging step to update only the corresponding expert policy parameters between two merging steps. Let $\mathbb{S}_t$ denote the set of indices for the selected objectives at step $t$. The update rule is defined as follows:

$$\boldsymbol{\theta}_{i,t+1} = \begin{cases} \boldsymbol{\theta}_{i,t} - \eta \nabla_{\boldsymbol{\theta}} \mathcal{L}_i(\pi_{\boldsymbol{\theta}_{i,t}}), & \text{if } t \bmod m \neq 0 \\ \boldsymbol{\rho}_t - \eta \nabla_{\boldsymbol{\rho}} \mathcal{L}_i(\pi_{\boldsymbol{\rho}_t}), & \text{if } t \bmod m = 0 \end{cases}, \forall i \in \mathbb{S}_t \tag{8}$$

where $t \bmod m$ denotes the remainder of $t$ divided by $m$. Note that when $t \bmod m \neq 0$, the subset remains unchanged, i.e., $\mathbb{S}_t = \mathbb{S}_{t-1}$. Various strategies can be used to merge the policy parameters $\boldsymbol{\theta}_{i,t}$ to compute $\boldsymbol{\rho}_t$. For simplicity, we adopt a linear combination:

$$\boldsymbol{\rho}_t = \sum_{i \in \mathbb{S}_t} \lambda_{i,t} \boldsymbol{\theta}_{i,t}, \text{ such that } \sum_{i \in \mathbb{S}_t} \lambda_{i,t} = 1, \forall t : t \bmod m = 0. \tag{9}$$

Furthermore, if $t \bmod m = 0$, a new subset of objectives $\mathbb{S}_t$ is selected by uniformly sampling $M$ objectives at random. The weights $\lambda_{i,t} \geq 0$ can be determined using Monte Carlo methods, by sampling different sets of coefficients and selecting the one that minimizes the weighted loss. To reduce computational overhead, a simpler alternative is to fix the weights as

$$\lambda_{i,t} = \frac{w_i}{\sum_{j \in \mathbb{S}_t} w_j} \tag{10}$$

aligning them with predefined objective preferences. Algorithm 1 summarizes the proposed algorithm. Since every $m$ steps involve a merging procedure similar to Rewarded Soups, we refer to the proposed method as *IterativeRS*, short for Iterative Rewarded Soups.

## 4.2 Analysis

This section analyzes the performance of IterativeRS. To gain a clearer understanding, we examine its convergence behavior in cases where the loss function $\mathcal{L}_i(\pi_{\boldsymbol{\theta}})$ is convex with respect to $\boldsymbol{\theta}$. While this

---

**Algorithm 1** IterativeRS: Iterative Multi-Objective Model Fine-Tuning

---
1: **Input:** Reference policy $\pi_{\text{ref}}$, learning rate $\eta$, merge frequency $m$.
2: Initialize $\pi_{\boldsymbol{\theta}_{i,1}}, \forall i \in \{1, \ldots, N\}$ as $\pi_{\text{ref}}$; $\mathbb{S}_0$ by sampling $M$ objectives uniformly.
3: **for** $t = 1, \ldots, T$ **do**
4:     Set $\mathbb{S}_t = \mathbb{S}_{t-1}$
5:     **if** $t \bmod m = 0$ **then**
6:         Merge policy weights $\{\boldsymbol{\theta}_{i,t}\}_{i=1}^{N}$ to obtain the shared parameter $\boldsymbol{\rho}_t$ as in equation 9.
7:         Sample uniformly at random $M$ objectives to update $\mathbb{S}_t$.
8:     **end if**
9:     For any objective $i \in \mathbb{S}_t$, update the policy parameter $\boldsymbol{\theta}_{i,t}$ as in equation 8.
10: **end for**
11: Merge all policy weights $\{\boldsymbol{\theta}_{i,T}\}_{i=1}^{N}$ to obtain the shared parameter $\boldsymbol{\rho}_T$.
12: **Output:** Policy $\pi_{\boldsymbol{\rho}_T}$.

---

convexity assumption may not hold in practical scenarios, the analysis provides valuable insight into the impact of hyperparameters on IterativeRS's performance. It is worth noting that MORLHF and Rewarded Soups can be viewed as special cases of IterativeRS by setting $\lambda_{i,t} = w_i$ and optimizing all objectives at each step. According to Algorithm 1 and Subsections 3.2 and 3.3, setting $m = 1$ in IterativeRS recovers MORLHF described by equation 4, while setting $m = T$ corresponds to Rewarded Soups.

The following assumptions are made for the analysis:

**A 1.** *Loss functions $\mathcal{L}_i(\cdot)$, $\forall i \in \{1, \ldots, N\}$ are $L$-smooth such that $\mathcal{L}_i(\pi_{\boldsymbol{\theta}_1}) \leq \mathcal{L}_i(\pi_{\boldsymbol{\theta}_2}) + \frac{L}{2}\|\boldsymbol{\theta}_1 - \boldsymbol{\theta}_2\|^2$, $\forall \boldsymbol{\theta}_1, \boldsymbol{\theta}_2$.*

**A 2.** *Loss functions $\mathcal{L}_i(\cdot)$, $\forall i \in \{1, \ldots, N\}$ are $\mu$-strongly convex such that $\mathcal{L}_i(\pi_{\boldsymbol{\theta}_1}) \geq \mathcal{L}_i(\pi_{\boldsymbol{\theta}_2}) + (\boldsymbol{\theta}_1 - \boldsymbol{\theta}_2)^\top \nabla \mathcal{L}_i(\boldsymbol{\theta}_2) + \frac{\mu}{2}\|\boldsymbol{\theta}_1 - \boldsymbol{\theta}_2\|^2$, $\forall \boldsymbol{\theta}_1, \boldsymbol{\theta}_2$.*

**A 3.** *Loss gradients are bounded from above as $\|\nabla\mathcal{L}_i(\pi_{\boldsymbol{\theta}})\| \leq G$, $\forall \boldsymbol{\theta}$, $\forall i \in \{1, \ldots, N\}$.*

Let the overall loss of a policy $\pi_{\boldsymbol{\theta}}$ be defined as

$$\mathcal{L}(\pi_{\boldsymbol{\theta}}) = \sum_{i=1}^{N} w_i \mathcal{L}_i(\pi_{\boldsymbol{\theta}}), \tag{11}$$

where $\mathcal{L}_i(\pi_{\boldsymbol{\theta}})$ is defined in equation 1. Let $\boldsymbol{\theta}^*$ denote the optimal policy parameters for the multi-objective loss, and let $\boldsymbol{\theta}_i^*$ denote the optimal policy parameters for the objective $i$, defined as

$$\boldsymbol{\theta}^* = \arg\min_{\boldsymbol{\theta}} \mathcal{L}(\pi_{\boldsymbol{\theta}}), \; \boldsymbol{\theta}_i^* = \arg\min_{\boldsymbol{\theta}} \mathcal{L}_i(\pi_{\boldsymbol{\theta}}) \tag{12}$$

The following theorem provides a convergence bound for IterativeRS, with the proof presented in the Appendix A. The theorem is proved under the assumption that the merged policy parameter is computed as $\boldsymbol{\rho}_t = \frac{N}{M}\sum_{i \in \mathbb{S}_t} w_i \boldsymbol{\theta}_{i,t}$ where $w_i = \frac{1}{N}$, $\forall i \in \{1, \ldots, N\}$. The extension to non-uniform weights $w_i$ is straightforward and is discussed in Appendix A.

**Theorem 1.** *Let the learning rate at step $t$ is set as $\eta_t = \frac{2}{\mu(\gamma+t)}$ where $\gamma = \max\{\frac{8L}{\mu}, m\} - 1$. Furthermore, let $\boldsymbol{\theta}_{ref}$ denote the policy parameter of the initial reference policy $\pi_{ref}$. Under assumptions A 1–A 3, the performance gap of policy learned by IterativeRS with respect to the optimal policy $\pi_{\boldsymbol{\theta}^*}$ is bounded from above as:*

$$\mathcal{L}(\pi_{\boldsymbol{\rho}_T}) - \mathcal{L}(\pi_{\boldsymbol{\theta}^*}) \leq \frac{4L}{\mu^2(\gamma+T)}\left(3L\Delta^* + 2(2(m-1)^2 + \frac{N-M}{N-1}\frac{m^2}{M})G^2\right)$$
$$+ \frac{\gamma L}{2(\gamma+T)}\|\boldsymbol{\theta}_{ref} - \boldsymbol{\theta}^*\|^2 \tag{13}$$

*where $\Delta^*$ be defined as:*

$$\Delta^* = \mathcal{L}(\pi_{\boldsymbol{\theta}^*}) - \sum_{i=1}^{N} w_i \mathcal{L}_i(\pi_{\boldsymbol{\theta}_i^*}). \tag{14}$$

In what follows, the effects of the hyperparameters are analyzed using Theorem 1. It is important to note, however, that a tighter performance gap upper bound in equation 13, does not necessarily translate to better performance during deployment. It primarily reflects improved convergence during training and may increase the risk of overfitting. Therefore, while the theoretical analysis helps in understanding the impact of hyperparameters, practical performance should be monitored using a validation set.

**Effects of $\pi_{\mathbf{ref}}$ and $\Delta^*$.** From equation 13, it can be inferred that decrease in $\|\boldsymbol{\theta}_{\mathrm{ref}} - \boldsymbol{\theta}^*\|$ improves the performance gap upper bound. This suggests that initializing with a stronger reference policy yields a more effective fine-tuned policy. Furthermore, equation 13 shows that smaller $\Delta^*$ results in tighter performance gap upper bound. A smaller $\Delta^*$ can be achieved when the optimal policies corresponding to individual objectives exhibit less variation. Therefore, Theorem 1 suggests that greater similarity among objectives facilitates learning the optimal policy.

**Choice of $M$ and $T$.** Using the bound in equation 13, it can be observed that increasing the number of selected objectives $M$ leads to a tighter upper bound on the performance gap. This is expected, as learning over a larger set of objectives at each time step typically results in a better final policy. However, increasing $M$ also increases the computational complexity. Similarly, equation 13 indicates that increasing the number of steps $T$ tightens the upper bound on the performance gap, but at the cost of greater computational complexity. Thus, a trade-off arises between minimizing the performance gap and managing computational cost.

**Choice of m.** In order to understand the effect of $m$ on the upper bound in equation 13, let break the upper bound into two terms $A_1$ and $A_2$ where

$$A_1 = \frac{12L\Delta^*}{\mu^2(\gamma + T)} \tag{15a}$$

$$A_2 = \frac{8L}{\mu^2(\gamma + T)} \left( 2(m-1)^2 + \frac{N - M}{N - 1} \frac{m^2}{M} \right) G^2 + \frac{\gamma L}{2(\gamma + T)} \|\boldsymbol{\theta}_{\mathrm{ref}} - \boldsymbol{\theta}^*\|^2. \tag{15b}$$

Given that $\gamma = \max\{\frac{8L}{\mu}, m\}$, if $m \geq \frac{8L}{\mu}$, increasing $m$ can lead to a reduction in the term $A_1$. On the other hand, increasing $m$ is more likely to increase the term $A_2$. The overall effect of $m$ on the upper bound depends on which term dominates. Therefore, the impact of $m$ is influenced by several factors, including the loss function and even the dataset, which may not be known a priori. As previously discussed, MORLHF and Rewarded Soups represent two extreme cases, where $m = 1$ (MORLHF) and $m = T$ (Rewarded Soups). However, a moderate choice of $m$ may yield the best trade-off. Therefore, it can be concluded that IterativeRS offers greater flexibility and potential for improvement by allowing arbitrary values of $m$.

## 5 Experiments

To evaluate the performance of IterativeRS, we conducted extensive experiments across a diverse set of tasks, including small molecule generation (Subsection 5.1), DNA sequence generation (Subsection 5.2), and text summarization (Subsection 5.3). We compare IterativeRS against state-of-the-art baselines: MORLHF [31], Rewarded Soups (RS) [42], and Rewards-in-Context (RiC) [52]. We assume that all objectives are equally important across all tasks, setting the weights to $w_1 = w_2 = w_3 = \frac{1}{3}$. It should be noted that the implementation of the MORLHF baseline in this section differs from the formulation presented in equation 4 in Subsection 3.2. To fine-tune models using IterativeRS, RS, and MORLHF, we employed PPO [46]. We evaluate the performance of each algorithm using the average rewards of its generated samples, both per objective and across all objectives. In addition, we report an *inverse coefficient of variation (ICV)* score to quantify performance consistency across objectives. For a given sample, the ICV score is defined as the average reward across all objectives divided by the standard deviation of those rewards. The average ICV score over $S$ samples is computed as

$$\text{ICV} = \frac{1}{S} \sum_{j=1}^{S} \frac{\frac{1}{N}(R_{j,1} + \ldots + R_{j,N})}{\text{std}(R_{j,1}, \ldots, R_{j,N})} \tag{16}$$

where $R_{j,i}$ denotes the reward obtained by sample $j$ on objective $i$, and $\text{std}(R_{j,1}, \ldots, R_{j,N})$ represents the standard deviation of rewards across objectives. A higher ICV score indicates lower variability

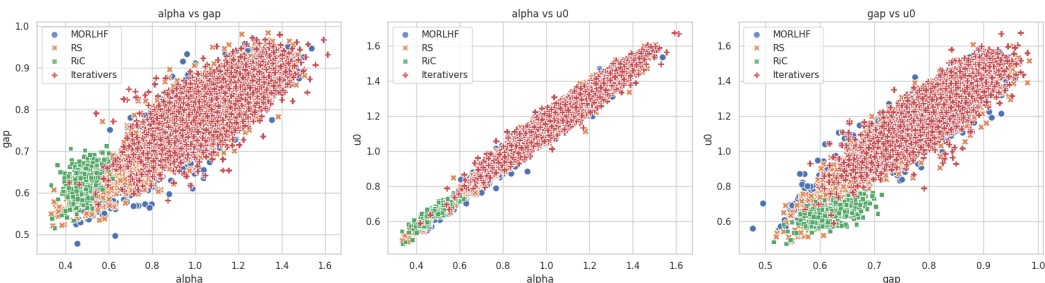

Figure 1: Pairwise scatter plots of generated molecules in the reward space for the three objectives.

Table 1: Average performance of Pareto-optimal molecules generated by multi-objective approaches.

|  | $\alpha$ **energy** | **gap** | $U_0$ **energy** | **Avg Reward** | **ICV** |
|---|---|---|---|---|---|
| MORLHF | 1.4229 | 0.9355 | 1.5146 | 1.2910 | 4.1883 |
| RS | 1.4134 | **0.9589** | 1.5464 | 1.3062 | **4.2674** |
| RiC | 0.5955 | 0.6795 | 0.7544 | 0.6765 | 3.7538 |
| IterativeRS | **1.5893** | 0.9508 | **1.6649** | **1.4017** | 3.5854 |

and more balanced performance across objectives. Codes are available at `https://github.com/pouyamghari/IterativeRS`.

## 5.1 Small Molecule Generation

The goal of this task is to generate small molecules that exhibit specific desirable energy properties. Specifically, the task involves generating molecules that (1) maximize polarizability ($\alpha$ energy), (2) maintain a moderate HOMO-LUMO gap, and (3) minimize internal energy at 0 K ($U_0$). To evaluate the properties of molecules generated by IterativeRS and the baseline methods, we use PAMNet [59] as the oracle model. PAMNet is specifically trained to predict molecular properties from the QM9 dataset. A GPT-2 model is pre-trained on SMILES representations of 2 million molecules from the MOSES dataset [40], resulting in a model referred to as MolGPT-2. This pre-trained model is then fine-tuned on the QM9 dataset [6, 45] to optimize for multiple objectives. To fine-tune models using IterativeRS, RS, and MORLHF, we employed PPO [46] with a reward model trained on the QM9 dataset. Rewards for each objective are normalized to the interval $[0, 1]$ using statistics computed from the training data. For RiC, supervised fine-tuning was performed using the QM9 dataset. To generate molecules, we first sample 10,000 SMILES representations using the fine-tuned model. Then, using RDKit, we construct 3D structures for each generated SMILES. Due to potential randomization in the 3D coordinates produced by RDKit, we generate 10 distinct 3D conformations for each SMILES. The resulting structures are then evaluated using PAMNet. More implementation details can be found in Appendix B.1.

Table 1 presents the performance of IterativeRS and other baseline methods in molecule generation. For IterativeRS, the merging frequency is set to $m = 4$. Moreover, for all reinforcement learning-based approaches (IterativeRS, RS, and MORLHF) the number of optimization steps is set to $T = 100$. To evaluate each method, we compute the Pareto front of the generated molecules and report the average reward for each objective within that front. If one generated molecule outperforms another (both produced by the same model) across all objectives, the former is said to dominate the latter and is included in the Pareto optimal set, while the dominated sample is considered suboptimal. As shown in Table 1, RL-based methods outperform RiC in terms of reward. This is likely because IterativeRS, RS, and MORLHF allow the pre-trained foundation model to interact with reward models during training, enabling it to explore and learn to generate higher-quality SMILES. In contrast, RiC relies solely on labeled data and lacks the exploration benefits provided by reinforcement learning. Since the distribution of pre-trained data differs from the labeled dataset, RL-based methods are better equipped to discover molecules with higher rewards than those present in the training set. Furthermore, as can be seen from Table 1, IterativeRS outperforms both MORLHF and RS in terms of average reward.

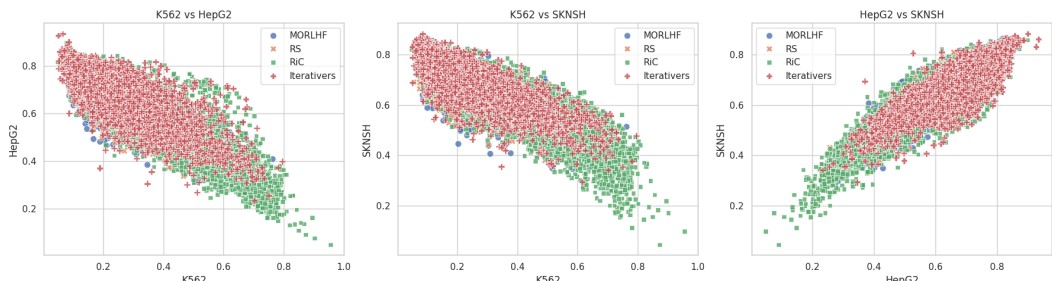

Figure 2: Pairwise scatter plots of generated DNA sequences in the reward space for the three objectives.

Table 2: Average performance of Pareto-optimal DNA sequences generated by multi-objective approaches.

|  | K562 | HepG2 | SKNSH | Avg Reward | ICV |
|---|---|---|---|---|---|
| MORLHF | 0.2724 | 0.7096 | 0.7183 | 0.5667 | 3.1356 |
| RS | 0.3057 | 0.6808 | 0.7131 | 0.5666 | 3.8235 |
| RiC | **0.4221** | 0.6615 | 0.6688 | 0.5842 | 2.4672 |
| IterativeRS | 0.3032 | **0.7370** | **0.7378** | **0.5927** | **3.8310** |

Figure 1 presents scatter plots of the molecules generated by each method. Each subplot depicts the relationship between two objectives, with each point representing a molecule generated by the corresponding model. These plots illustrate how the generated molecules are distributed across the objective space. Notably, Figure 1 shows that the highest-scoring molecules are produced by IterativeRS. This is particularly important for molecule design, where the goal is often to identify a small number of molecules with optimal properties. These results highlight the effectiveness of IterativeRS in the small molecule generation task.

## 5.2 DNA Sequence Generation

The goal is to generate DNA sequences that exhibit desired regulatory activities in specific cell lines K562, HepG2 and SKNSH. To this end, a GPT-2 model referred to as DNAGPT-2 is pre-trained on approximately 700,000 unlabeled DNA sequences, each 200 base pairs long, from the MPRA dataset [18], comprising over 35 million tokens. The objective is to generate sequences with maximal regulatory activity across three different cell lines. For fine-tuning, we use a labeled subset of 100,000 sequences along with their corresponding activity measurements in the three target cell lines. To assess the quality of the generated sequences, we utilize the Malinois model [18] as an oracle predictor of regulatory activity. Rewards for each objective are normalized to the interval $[0, 1]$ using statistics computed from the training data. Each method generates 10,000 DNA sequences. More implementation details can be found in Appendix B.2.

Table 2 presents the performance of different algorithms in generating DNA sequences. For IterativeRS, the merging frequency is set to $m = 8$, and for all reinforcement learning (RL)-based methods the number of optimization steps is fixed at $T = 200$. For each method, the Pareto front of the generated DNA sequences is extracted based on their rewards across the objectives. As shown in Table 2, RiC achieves higher average reward scores than MORLHF and RS. Unlike the small molecule generation task, the distribution of data used to pre-train the foundation model aligns closely with the supervised training data for DNA sequences. As a result, RL-based methods provide less benefit in this setting compared to supervised fine-tuning. While IterativeRS achieves an average reward that is 1% higher than RiC, IterativeRS attains a 35% higher ICV score, indicating significantly greater consistency in performance across objectives. Moreover, IterativeRS outperforms both RS and MORLHF in terms of average reward.

Figure 2 presents scatter plots of DNA sequences generated by each method, with each subplot comparing the rewards of two objectives. The results indicate that sequences generated by RiC exhibit greater variability across objectives compared to those produced by IterativeRS. Notably,

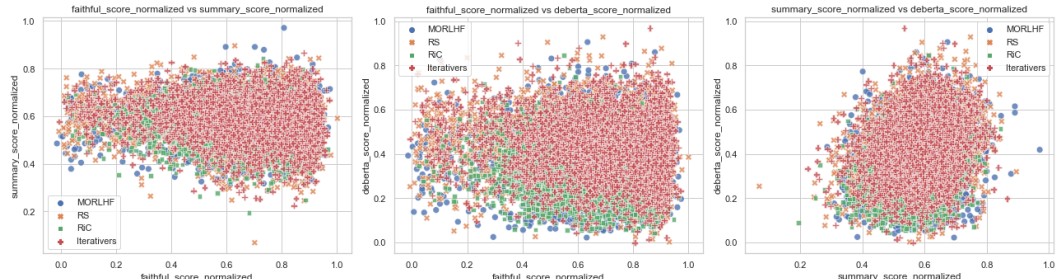

Figure 3: Pairwise scatter plots of generated summaries in the reward space for the three objectives.

Table 3: Average performance of text summarization by multi-objective approaches.

|  | faithful | summary | deberta | Avg Score | ICV |
|---|---|---|---|---|---|
| MORLHF | 0.6530 | 0.5778 | 0.3857 | 0.4525 | 4.5500 |
| RS | 0.6732 | 0.5807 | 0.4296 | 0.4732 | 4.5870 |
| RiC | 0.6497 | 0.5688 | 0.3455 | 0.4518 | 3.9579 |
| IterativeRS | **0.6927** | **0.5854** | **0.4398** | **0.4849** | **4.9134** |

IterativeRS generates fewer DNA sequences with low rewards, demonstrating a more consistent performance compared to RiC.

## 5.3 Text Summarization

The task is to summarize Reddit posts. To accomplish this, we use Llama-3.2-3B-Instruct as the base model. This foundation model is fine-tuned on the Reddit Summary dataset [49] for the post summarization task. To evaluate the quality of the generated summaries, we employ three different reward models: bart-faithful-summary [9], gpt2-reward-summary [2], deberta-v3 [3]. The rewards assigned by bart-faithful-summary, gpt2-reward-summary, deberta-v3 are referred to as the *faithful* score, *summary* score, and *deberta* score, respectively. All reported rewards are normalized to the range $[0, 1]$ using statistics computed from the training dataset. The merging steps in RS and IterativeRS are performed using seven different sets of merging weights. For each set, a merged model is obtained, and the model that achieves the highest average reward according to the reward models is selected as the final merged model for the text summarization task. It is worth noting that one of the main differences between IterativeRS and RS is that, according to Algorithm 1, IterativeRS performs merging both during and after training, whereas RS merges the expert policies only once after training. More implementation details can be found in Appendix B.3.

Table 3 presents the performance of the algorithms on the text summarization task for Reddit posts. The merging frequency for IterativeRS is set to $m = 40$, while the number of steps for all RL-based methods is $T = 160$. For each generated summary, we computed the average of the faithful, summary, deberta, and ROUGE scores as the evaluation metric to incorporate a standard metric such as ROUGE in addition to the scores assigned by the reward models. This average is reported as *Avg Score* in the table. The ICV score is calculated using the faithful, summary, and deberta reward scores. The results in Table 3 show that IterativeRS outperforms the other baselines across all metrics. These findings indicate that employing IterativeRS can lead to improvements over RL-based approaches such as MORLHF and RS. It is also worth noting that RiC is a supervised fine-tuning (SFT) approach, unlike the RL-based methods. Although IterativeRS achieves higher scores than RiC, the superiority of RL-based approaches over SFT methods such as RiC is not universally generalizable and is influenced by the experimental conditions. Figure 3 shows scatter plots of the summaries generated by each method, with each subplot comparing two objectives. As seen in the figure, IterativeRS is less likely to produce responses with relatively low scores.

---

[2] https://huggingface.co/Tristan/gpt2_reward_summarization
[3] https://huggingface.co/OpenAssistant/reward-model-deberta-v3-large-v2

# 6 Conclusion

This paper introduced IterativeRS, an iterative multi-objective reinforcement learning algorithm for fine-tuning foundation models. IterativeRS fine-tunes a separate model for each objective to capture objective-specific knowledge, while mitigating divergence across expert models through an iterative merge-and-fine-tune strategy. The paper presents a theoretical analysis of the convergence properties of IterativeRS, offering deeper insight into its behavior. Furthermore, by formulating the problem as an optimization task, our work can potentially open new directions for improving multi-objective fine-tuning of foundation models. Experimental results across diverse tasks including small molecule generation, DNA sequence generation, and text summarization demonstrate that IterativeRS achieves higher average rewards compared to both MORLHF and Rewarded Soups.

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

# A Proof of Theorem 1

This section proves Theorem 1. The notations used in this section are summarized in Table 4. For simplicity of analysis, we assume that all data samples are used at each step, although the case where a random subset of data is sampled at each step can also be considered. In that case, Assumption A 3 can be modified to $\mathbb{E}[\|\nabla \mathcal{L}_i(\pi_{\boldsymbol{\theta}})\|] \leq G$, where the expectation is taken with respect to the randomness in data sampling. Extending the results to the case with random data sampling is straightforward. Let $\boldsymbol{\psi}_t$ be defined as $\boldsymbol{\psi}_t = \sum_{i=1}^{N} w_i \boldsymbol{\theta}_{i,t}$. Furthermore, let the merged policy parameter $\boldsymbol{\rho}_t$ be defined as

$$\boldsymbol{\rho}_t = \begin{cases} \frac{N}{M} \sum_{i \in \mathbb{S}_t} w_i \boldsymbol{\theta}_{i,t}, & \text{if } t \bmod m = 0 \\ \boldsymbol{\psi}_t, & \text{if } t \bmod m \neq 0 \end{cases} \tag{17}$$

We can write

$$\|\boldsymbol{\rho}_{t+1} - \boldsymbol{\theta}^*\|^2 = \|\boldsymbol{\rho}_{t+1} - \boldsymbol{\psi}_{t+1} + \boldsymbol{\psi}_{t+1} - \boldsymbol{\theta}^*\|^2$$
$$= \|\boldsymbol{\rho}_{t+1} - \boldsymbol{\psi}_{t+1}\|^2 + 2(\boldsymbol{\rho}_{t+1} - \boldsymbol{\psi}_{t+1})^{\top} (\boldsymbol{\psi}_{t+1} - \boldsymbol{\theta}^*) + \|\boldsymbol{\psi}_{t+1} - \boldsymbol{\theta}^*\|^2. \tag{18}$$

To obtain an upper bound on $\|\boldsymbol{\psi}_{t+1} - \boldsymbol{\theta}^*\|^2$, consider the Lemma 1. This lemma is taken from [33].

**Lemma 1.** *Suppose Assumptions A 1 and A 2 hold. If $\eta_t \leq \frac{1}{4L}$, then the following inequality holds:*

$$\|\boldsymbol{\psi}_{t+1} - \boldsymbol{\theta}^*\|^2 \leq (1 - \eta_t \mu) \|\boldsymbol{\psi}_t - \boldsymbol{\theta}^*\|^2 + 6L\eta_t^2 \Delta^* + 2 \sum_{i=1}^{N} w_i \|\boldsymbol{\psi}_t - \boldsymbol{\theta}_{i,t}\|^2 \tag{19}$$

*where $\Delta^* = \mathcal{L}(\pi_{\boldsymbol{\theta}^*}) - \sum_{i=1}^{N} w_i \mathcal{L}_i(\pi_{\boldsymbol{\theta}_i^*})$.*

*Proof.* Let $\boldsymbol{g}_t$ be defined as $\boldsymbol{g}_t := \sum_{i=1}^{N} w_i \nabla \mathcal{L}_i(\pi_{\boldsymbol{\theta}_{i,t}})$. It can be inferred that $\boldsymbol{\psi}_{t+1} = \boldsymbol{\psi}_t - \eta_t \boldsymbol{g}_t$. Therefore, we may write

$$\|\boldsymbol{\psi}_{t+1} - \boldsymbol{\theta}^*\|^2 = \|\boldsymbol{\psi}_t - \eta_t \boldsymbol{g}_t - \boldsymbol{\theta}^*\|^2 = \|\boldsymbol{\psi}_t - \boldsymbol{\theta}^*\|^2 - 2\eta_t \boldsymbol{g}_t^{\top} (\boldsymbol{\psi}_t - \boldsymbol{\theta}^*) + \eta_t^2 \|\boldsymbol{g}_t\|^2. \tag{20}$$

From $L$-smoothness of $\mathcal{L}_i$ stated in assumption A 1, it follows that

$$\|\nabla \mathcal{L}_i(\pi_{\boldsymbol{\theta}_{i,t}})\|^2 \leq 2L \left( \mathcal{L}_i(\pi_{\boldsymbol{\theta}_{i,t}}) - \mathcal{L}_i(\pi_{\boldsymbol{\theta}_i^*}) \right) \tag{21}$$

where $\boldsymbol{\theta}_i^* = \arg\min_{\boldsymbol{\theta}} \mathcal{L}_i(\pi_{\boldsymbol{\theta}})$. Using the above inequality and due to the convexity of $\|\cdot\|^2$, we can write

$$\eta_t^2 \|\boldsymbol{g}_t\|^2 \leq \eta_t^2 \sum_{i=1}^{N} w_i \|\nabla \mathcal{L}_i(\pi_{\boldsymbol{\theta}_{i,t}})\|^2 \leq 2\eta_t^2 L \sum_{i=1}^{N} w_i \left( \mathcal{L}_i(\pi_{\boldsymbol{\theta}_{i,t}}) - \mathcal{L}_i(\pi_{\boldsymbol{\theta}_i^*}) \right). \tag{22}$$

Furthermore, we can rewrite the term $-2\eta_t \boldsymbol{g}_t^{\top} (\boldsymbol{\psi}_t - \boldsymbol{\theta}^*)$ in equation 20 as

$$-2\eta_t \boldsymbol{g}_t^{\top} (\boldsymbol{\psi}_t - \boldsymbol{\theta}^*) = -2\eta_t \sum_{i=1}^{N} w_i \nabla \mathcal{L}_i(\pi_{\boldsymbol{\theta}_{i,t}})^{\top} (\boldsymbol{\psi}_t - \boldsymbol{\theta}_{i,t})$$
$$-2\eta_t \sum_{i=1}^{N} w_i \nabla \mathcal{L}_i(\pi_{\boldsymbol{\theta}_{i,t}})^{\top} (\boldsymbol{\theta}_{i,t} - \boldsymbol{\theta}^*) \tag{23}$$

Using AM-GM inequality, we can obtain

$$-2\nabla \mathcal{L}_i(\pi_{\boldsymbol{\theta}_{i,t}})^{\top} (\boldsymbol{\psi}_t - \boldsymbol{\theta}_{i,t}) \leq \frac{1}{\eta_t} \|\boldsymbol{\psi}_t - \boldsymbol{\theta}_{i,t}\|^2 + \eta_t \|\nabla \mathcal{L}_i(\pi_{\boldsymbol{\theta}_{i,t}})\|^2, \tag{24}$$

while due to $\mu$-strong convexity of $\mathcal{L}_i$, we can write

$$-\nabla \mathcal{L}_i(\pi_{\boldsymbol{\theta}_{i,t}})^{\top} (\boldsymbol{\theta}_{i,t} - \boldsymbol{\theta}^*) \leq -(\mathcal{L}_i(\pi_{\boldsymbol{\theta}_{i,t}}) - \mathcal{L}_i(\pi_{\boldsymbol{\theta}^*})) - \frac{\mu}{2} \|\boldsymbol{\theta}_{i,t} - \boldsymbol{\theta}^*\|^2 \tag{25}$$

Taking a weighted average over the objectives and applying Jensen's inequality to the right-hand side of equation 25, we obtain

$$-\sum_{i=1}^{N} w_i \nabla \mathcal{L}_i(\pi_{\boldsymbol{\theta}_{i,t}})^{\top} (\boldsymbol{\theta}_{i,t} - \boldsymbol{\theta}^*) \leq -\sum_{i=1}^{N} w_i (\mathcal{L}_i(\pi_{\boldsymbol{\theta}_{i,t}}) - \mathcal{L}_i(\pi_{\boldsymbol{\theta}^*})) - \frac{\mu}{2} \|\boldsymbol{\psi}_t - \boldsymbol{\theta}^*\|^2. \tag{26}$$

Table 4: Notation Table.

| Symbol | Description |
|---|---|
| $N$ | Number of objectives |
| $M$ | Number of randomly selected objectives at each step $t$ |
| $\mathbb{S}_t$ | Set of selected objectives at step $t$ |
| $w_i$ | Preference weight associated with the $i$-th objective |
| $\pi_{\boldsymbol{\theta}}$ | Policy of the language model parameterized by $\boldsymbol{\theta}$ |
| $\boldsymbol{\theta}_{i,t}$ | Parameters of the policy associated with the $i$-th objective at optimization step $t$ |
| $\boldsymbol{\rho}_t$ | Merged parameters of all policies at step $t$, defined as in equation 17 |
| $\boldsymbol{\psi}_t$ | Weighted average of policy parameters, defined as $\boldsymbol{\psi}_t = \sum_{i=1}^{N} w_i \boldsymbol{\theta}_{i,t}$ |
| $\boldsymbol{g}_t$ | Weighted average of policy gradients, defined as $\boldsymbol{g}_t = \sum_{i=1}^{N} w_i \nabla \mathcal{L}_i(\pi_{\boldsymbol{\theta}_{i,t}})$ |
| $\boldsymbol{\theta}^*$ | Optimal policy parameter for the multi-objective loss, defined as in equation 12 |
| $\boldsymbol{\theta}_i^*$ | Optimal policy parameter for objective $i$, defined as in equation 12 |

Furthermore, taking a weighted average and applying the inequality in equation 21 to equation 24, we get

$$-2\sum_{i=1}^{N} w_i \nabla \mathcal{L}_i(\pi_{\boldsymbol{\theta}_{i,t}})^\top (\boldsymbol{\psi}_t - \boldsymbol{\theta}_{i,t}) \leq \sum_{i=1}^{N} \frac{w_i}{\eta_t} \|\boldsymbol{\psi}_t - \boldsymbol{\theta}_{i,t}\|^2$$
$$+ 2\eta_t L \sum_{i=1}^{N} w_i \left( \mathcal{L}_i(\pi_{\boldsymbol{\theta}_{i,t}}) - \mathcal{L}_i(\pi_{\boldsymbol{\theta}_i^*}) \right). \quad (27)$$

Combining equation 20 with equation 22, equation 23, equation 26 and equation 27, we arrive at

$$\|\boldsymbol{\psi}_{t+1} - \boldsymbol{\theta}^*\|^2 \leq (1 - \eta_t \mu)\|\boldsymbol{\psi}_t - \boldsymbol{\theta}^*\|^2 + \sum_{i=1}^{N} w_i \|\boldsymbol{\psi}_t - \boldsymbol{\theta}_{i,t}\|^2$$
$$+ 4\eta_t^2 L \sum_{i=1}^{N} w_i \left( \mathcal{L}_i(\pi_{\boldsymbol{\theta}_{i,t}}) - \mathcal{L}_i(\pi_{\boldsymbol{\theta}_i^*}) \right) - 2\eta_t \sum_{i=1}^{N} w_i (\mathcal{L}_i(\pi_{\boldsymbol{\theta}_{i,t}}) - \mathcal{L}_i(\pi_{\boldsymbol{\theta}^*})) \quad (28)$$

Taking the definition of $\Delta^*$ into account, the last two terms in the right hand side of equation 28 can be rewritten as

$$4\eta_t^2 L \sum_{i=1}^{N} w_i \left( \mathcal{L}_i(\pi_{\boldsymbol{\theta}_{i,t}}) - \mathcal{L}_i(\pi_{\boldsymbol{\theta}_i^*}) \right) - 2\eta_t \sum_{i=1}^{N} w_i (\mathcal{L}_i(\pi_{\boldsymbol{\theta}_{i,t}}) - \mathcal{L}_i(\pi_{\boldsymbol{\theta}^*}))$$
$$= -2\eta_t (1 - 2\eta_t L) \sum_{i=1}^{N} w_i \left( \mathcal{L}_i(\pi_{\boldsymbol{\theta}_{i,t}}) - \mathcal{L}_i(\pi_{\boldsymbol{\theta}_i^*}) \right) + 2\eta_t \sum_{i=1}^{N} w_i (\mathcal{L}_i(\pi_{\boldsymbol{\theta}^*}) - \mathcal{L}_i(\pi_{\boldsymbol{\theta}_i^*}))$$
$$= -2\eta_t (1 - 2\eta_t L) \sum_{i=1}^{N} w_i \left( \mathcal{L}_i(\pi_{\boldsymbol{\theta}_{i,t}}) - \mathcal{L}_i(\pi_{\boldsymbol{\theta}^*}) \right) + 4\eta_t^2 L \sum_{i=1}^{N} w_i (\mathcal{L}_i(\pi_{\boldsymbol{\theta}^*}) - \mathcal{L}_i(\pi_{\boldsymbol{\theta}_i^*}))$$
$$= -2\eta_t (1 - 2\eta_t L) \sum_{i=1}^{N} w_i \left( \mathcal{L}_i(\pi_{\boldsymbol{\theta}_{i,t}}) - \mathcal{L}_i(\pi_{\boldsymbol{\theta}^*}) \right) + 4\eta_t^2 L \Delta^*. \quad (29)$$

To bound $\sum_{i=1}^{N} w_i \left( \mathcal{L}_i(\pi_{\boldsymbol{\theta}_{i,t}}) - \mathcal{L}_i(\pi_{\boldsymbol{\theta}^*}) \right)$, considering the convexity of $\mathcal{L}_i$, we can write

$$\sum_{i=1}^{N} w_i \left( \mathcal{L}_i(\pi_{\boldsymbol{\theta}_{i,t}}) - \mathcal{L}_i(\pi_{\boldsymbol{\theta}^*}) \right) = \sum_{i=1}^{N} w_i \left( \mathcal{L}_i(\pi_{\boldsymbol{\theta}_{i,t}}) - \mathcal{L}_i(\pi_{\boldsymbol{\psi}_t}) \right) + \sum_{i=1}^{N} w_i \left( \mathcal{L}_i(\pi_{\boldsymbol{\psi}_t}) - \mathcal{L}_i(\pi_{\boldsymbol{\theta}^*}) \right)$$
$$\geq \sum_{i=1}^{N} w_i \nabla \mathcal{L}_i(\boldsymbol{\psi}_t)^\top (\boldsymbol{\theta}_{i,t} - \boldsymbol{\psi}_t) + \mathcal{L}_i(\pi_{\boldsymbol{\psi}_t}) - \mathcal{L}_i(\pi_{\boldsymbol{\theta}^*}). \quad (30)$$

Applying AM-GM inequality to the right hand side of equation 30, we get

$$\sum_{i=1}^{N} w_i \left( \mathcal{L}_i(\pi_{\boldsymbol{\theta}_{i,t}}) - \mathcal{L}_i(\pi_{\boldsymbol{\theta}^*}) \right) \geq \sum_{i=1}^{N} -\frac{w_i}{2} \left( \eta_t \|\nabla \mathcal{L}_i(\pi_{\boldsymbol{\psi}_t})\|^2 + \frac{1}{\eta_t} \|\boldsymbol{\theta}_{i,t} - \boldsymbol{\psi}_t\|^2 \right)$$
$$+ \mathcal{L}_i(\pi_{\boldsymbol{\psi}_t}) - \mathcal{L}_i(\pi_{\boldsymbol{\theta}^*}) \tag{31}$$

Applying the inequality in equation 21 to the right hand side of equation 31, we conclude that

$$\sum_{i=1}^{N} w_i \left( \mathcal{L}_i(\pi_{\boldsymbol{\theta}_{i,t}}) - \mathcal{L}_i(\pi_{\boldsymbol{\theta}^*}) \right) \geq -\sum_{i=1}^{N} w_i \left( \eta_t L \left( \mathcal{L}_i(\pi_{\boldsymbol{\psi}_t}) - \mathcal{L}_i(\pi_{\boldsymbol{\theta}_i^*}) \right) + \frac{1}{2\eta_t} \|\boldsymbol{\theta}_{i,t} - \boldsymbol{\psi}_t\|^2 \right)$$
$$+ \mathcal{L}_i(\pi_{\boldsymbol{\psi}_t}) - \mathcal{L}_i(\pi_{\boldsymbol{\theta}^*}). \tag{32}$$

Due the fact that $0 \leq \eta_t \leq \frac{1}{4L}$, it can be concluded that $\eta_t \leq 2\eta_t(1 - 2\eta_t L) \leq 2\eta_t$. Multiplying both sides of equation 32 by $-2\eta_t(1 - 2\eta_t L)$, we obtain

$$- 2\eta_t(1 - 2\eta_t L) \sum_{i=1}^{N} w_i \left( \mathcal{L}_i(\pi_{\boldsymbol{\theta}_{i,t}}) - \mathcal{L}_i(\pi_{\boldsymbol{\theta}^*}) \right)$$
$$\leq \sum_{i=1}^{N} w_i \left( 2\eta_t^2(1 - 2\eta_t L)L \left( \mathcal{L}_i(\pi_{\boldsymbol{\psi}_t}) - \mathcal{L}_i(\pi_{\boldsymbol{\theta}_i^*}) \right) + (1 - 2\eta_t L)\|\boldsymbol{\theta}_{i,t} - \boldsymbol{\psi}_t\|^2 \right)$$
$$- 2\eta_t(1 - 2\eta_t L) \left( \mathcal{L}_i(\pi_{\boldsymbol{\psi}_t}) - \mathcal{L}_i(\pi_{\boldsymbol{\theta}^*}) \right). \tag{33}$$

The inequality in equation 33 can be rewritten as

$$- 2\eta_t(1 - 2\eta_t L) \sum_{i=1}^{N} w_i \left( \mathcal{L}_i(\pi_{\boldsymbol{\theta}_{i,t}}) - \mathcal{L}_i(\pi_{\boldsymbol{\theta}^*}) \right)$$
$$\leq 2\eta_t^2(1 - 2\eta_t L)L\Delta^* + \sum_{i=1}^{N} w_i(1 - 2\eta_t L)\|\boldsymbol{\theta}_{i,t} - \boldsymbol{\psi}_t\|^2$$
$$+ (\eta_t L - 1)2\eta_t(1 - 2\eta_t L) \left( \mathcal{L}_i(\pi_{\boldsymbol{\psi}_t}) - \mathcal{L}_i(\pi_{\boldsymbol{\theta}^*}) \right). \tag{34}$$

Using the facts that $\mathcal{L}_i(\pi_{\boldsymbol{\psi}_t}) - \mathcal{L}_i(\pi_{\boldsymbol{\theta}^*}) \geq 0$, $\eta_t L - 1 \leq -\frac{3}{4}$ and $1 - 2\eta_t L \leq 1$, from equation 34 we obtain

$$-2\eta_t(1 - 2\eta_t L) \sum_{i=1}^{N} w_i \left( \mathcal{L}_i(\pi_{\boldsymbol{\theta}_{i,t}}) - \mathcal{L}_i(\pi_{\boldsymbol{\theta}^*}) \right) \leq 2\eta_t^2 L\Delta^* + \sum_{i=1}^{N} w_i \|\boldsymbol{\theta}_{i,t} - \boldsymbol{\psi}_t\|^2. \tag{35}$$

Combining equation 28 with equation 29 and equation 35 proves the Lemma. $\qquad \square$

Since IterativeRS merges every $m$ steps, there exists $t'$ such that $t - t' < m$ and $\boldsymbol{\theta}_{i,t'} = \boldsymbol{\psi}_{t'}$, $\forall i \in \{1, \dots, N\}$. Considering the facts that $\boldsymbol{\psi}_{t'}$ is the expected value of $\{\boldsymbol{\theta}_{i,t'}\}_{i=1}^{N}$, over distribution $\{w_i\}_{i=1}^{N}$ and $\mathbb{E}\|X - \mathbb{E}[X]\|^2 \leq \|\mathbb{E}[X]\|^2$, we can conclude that

$$\sum_{i=1}^{N} w_i \|\boldsymbol{\psi}_t - \boldsymbol{\theta}_{i,t}\|^2 = \sum_{i=1}^{N} w_i \|\boldsymbol{\psi}_t - \boldsymbol{\psi}_{t'} + \boldsymbol{\theta}_{i,t'} - \boldsymbol{\theta}_{i,t}\|^2 \leq \sum_{i=1}^{N} w_i \|\boldsymbol{\theta}_{i,t'} - \boldsymbol{\theta}_{i,t}\|^2. \tag{36}$$

Assume that $\eta_t$ is selected such that it is non-increasing and satisfies $\eta_t \leq 2\eta_{t+m}$, $\forall t$. Taking the assumption A 3 into account, we can infer that

$$\sum_{i=1}^{N} w_i \|\boldsymbol{\theta}_{i,t'} - \boldsymbol{\theta}_{i,t}\|^2 = \sum_{i=1}^{N} w_i \| \sum_{\tau=0}^{t-t'} \eta_{t'+\tau} \nabla \mathcal{L}_i(\pi_{\boldsymbol{\theta}_{i,t'}})\|^2 \leq 4(m-1)^2 \eta_t^2 G^2. \tag{37}$$

Combining equation 37 with equation 19, we get

$$\|\boldsymbol{\psi}_{t+1} - \boldsymbol{\theta}^*\|^2 \leq (1 - \eta_t \mu)\|\boldsymbol{\psi}_t - \boldsymbol{\theta}^*\|^2 + 6L\eta_t^2 \Delta^* + 8(m-1)^2 \eta_t^2 G^2. \tag{38}$$

Recall that IterativeRS in Algorithm 1 samples $M$ objectives uniformly without replacement. Such a sampling scheme is unbiased and we can write

$$\mathbb{E}_{\mathbb{S}_t}[\boldsymbol{\rho}_{t+1}] = \boldsymbol{\psi}_{t+1} \tag{39}$$

where $\mathbb{E}_{\mathbb{S}_t}[\cdot]$ denote the expectation with respect to sampling randomization. Therefore, taking expectation from equation 18 leads to

$$\mathbb{E}_{\mathbb{S}_t}[\|\boldsymbol{\rho}_{t+1} - \boldsymbol{\theta}^*\|^2] = \mathbb{E}_{\mathbb{S}_t}[\|\boldsymbol{\rho}_{t+1} - \boldsymbol{\psi}_{t+1}\|^2] + \|\boldsymbol{\psi}_{t+1} - \boldsymbol{\theta}^*\|^2. \tag{40}$$

Combining equation 40 with equation 38, we get

$$\begin{aligned}
\mathbb{E}_{\mathbb{S}_t}[\|\boldsymbol{\rho}_{t+1} - \boldsymbol{\theta}^*\|^2] \leq &\mathbb{E}_{\mathbb{S}_t}[\|\boldsymbol{\rho}_{t+1} - \boldsymbol{\psi}_{t+1}\|^2] \\
&+ (1 - \eta_t\mu)\|\boldsymbol{\psi}_t - \boldsymbol{\theta}^*\|^2 + 6L\eta_t^2\Delta^* + 8(m-1)^2\eta_t^2 G^2.
\end{aligned} \tag{41}$$

According to equation 17, if $(t+1) \bmod m \neq 0$, it can concluded that $\mathbb{E}_{\mathbb{S}_t}[\|\boldsymbol{\rho}_{t+1} - \boldsymbol{\psi}_{t+1}\|^2] = 0$. If $(t+1) \bmod m = 0$, considering the assumption that $w_1 = \ldots = w_N = \frac{1}{N}$, the term $\mathbb{E}_{\mathbb{S}_t}[\|\boldsymbol{\rho}_{t+1} - \boldsymbol{\psi}_{t+1}\|^2]$ can be expressed as:

$$\begin{aligned}
\mathbb{E}_{\mathbb{S}_t}[\|\boldsymbol{\rho}_{t+1} - \boldsymbol{\psi}_{t+1}\|^2] = &\mathbb{E}_{\mathbb{S}_t}\left\|\frac{1}{M}\sum_{i\in\mathbb{S}_{t+1}}\boldsymbol{\theta}_{i,t+1} - \boldsymbol{\psi}_{t+1}\right\|^2 \\
= &\frac{1}{M^2}\mathbb{E}_{\mathbb{S}_t}\left\|\sum_{i=1}^N \Pr[i\in\mathbb{S}_{t+1}](\boldsymbol{\theta}_{i,t+1} - \boldsymbol{\psi}_{t+1})\right\|^2 \\
= &\frac{1}{M^2}\sum_{i=1}^N \Pr[i\in\mathbb{S}_{t+1}]\|\boldsymbol{\theta}_{i,t+1} - \boldsymbol{\psi}_{t+1}\|^2 \\
&+ \frac{1}{M^2}\sum_{i\neq j}\Pr[i,j\in\mathbb{S}_{t+1}](\boldsymbol{\theta}_{i,t+1} - \boldsymbol{\psi}_{t+1})^\top(\boldsymbol{\theta}_{j,t+1} - \boldsymbol{\psi}_{t+1}).
\end{aligned} \tag{42}$$

Considering the facts that $\Pr[i\in\mathbb{S}_{t+1}] = \frac{M}{N}$ and $\Pr[i,j\in\mathbb{S}_{t+1}] = \frac{M(M-1)}{N(N-1)}$ and

$$\begin{aligned}
\|\sum_{i=1}^N \boldsymbol{\theta}_{i,t+1} - \boldsymbol{\psi}_{t+1}\|^2 = &\sum_{i=1}^N \|\boldsymbol{\theta}_{i,t+1} - \boldsymbol{\psi}_{t+1}\|^2 \\
&+ \sum_{i,j\in\mathbb{S}_{t+1}}(\boldsymbol{\theta}_{i,t+1} - \boldsymbol{\psi}_{t+1})^\top(\boldsymbol{\theta}_{j,t+1} - \boldsymbol{\psi}_{t+1}) = 0,
\end{aligned} \tag{43}$$

we can rewrite equation 42 as

$$\mathbb{E}_{\mathbb{S}_t}[\|\boldsymbol{\rho}_{t+1} - \boldsymbol{\psi}_{t+1}\|^2] = \frac{1}{M(N-1)}\left(1 - \frac{M}{N}\right)\sum_{i=1}^N \|\boldsymbol{\theta}_{i,t+1} - \boldsymbol{\psi}_{t+1}\|^2 \tag{44}$$

Using equation 36 and equation 37, from equation 44 we arrive at

$$\mathbb{E}_{\mathbb{S}_t}[\|\boldsymbol{\rho}_{t+1} - \boldsymbol{\psi}_{t+1}\|^2] \leq \frac{4N}{M(N-1)}\left(1 - \frac{M}{N}\right)\eta_t^2 m^2 G^2. \tag{45}$$

Combining equation 45 with equation 41, we get

$$\begin{aligned}
\mathbb{E}_{\mathbb{S}_t}[\|\boldsymbol{\rho}_{t+1} - \boldsymbol{\theta}^*\|^2] \leq &(1 - \eta_t\mu)\|\boldsymbol{\psi}_t - \boldsymbol{\theta}^*\|^2 + 6L\eta_t^2\Delta^* \\
&+ 4\left(2(m-1)^2 + \frac{N-M}{M(N-1)}m^2\right)\eta_t^2 G^2
\end{aligned} \tag{46}$$

Define $B$ as

$$B = 6L\Delta^* + 4\left(2(m-1)^2 + \frac{N-M}{M(N-1)}m^2\right)G^2. \tag{47}$$

With a step size chosen as $\eta_t = \frac{\beta}{t+\gamma}$ for some $\beta > \frac{1}{\mu}$ and $\gamma > 0$ satisfying $\eta_1 \leq \min\{\frac{1}{\mu}, \frac{1}{4L}\}$ and $\eta_t \leq 2\eta_{t+m}$, using induction it can be proved that $\mathbb{E}_{\mathbb{S}_t}[\|\boldsymbol{\rho}_t - \boldsymbol{\theta}^*\|^2] \leq \frac{v}{\gamma+t}$ where

$$v = \max\left\{\frac{\beta^2 B}{\beta\mu - 1}, (\gamma + 1)\|\boldsymbol{\psi}_1 - \boldsymbol{\theta}^*\|^2\right\}. \tag{48}$$

Since $\boldsymbol{\rho}_t$ is an unbiased estimator of $\boldsymbol{\psi}_t$, we can conclude that $\mathbb{E}_{\mathbb{S}_t}[\|\boldsymbol{\rho}_t - \boldsymbol{\theta}^*\|^2] = \|\boldsymbol{\psi}_t - \boldsymbol{\theta}^*\|^2$. Definition of $v$ ensures that $\|\boldsymbol{\psi}_t - \boldsymbol{\theta}^*\|^2 \leq \frac{v}{\gamma+t}$ for $t = 1$. Assume that $\|\boldsymbol{\psi}_t - \boldsymbol{\theta}^*\|^2 \leq \frac{v}{\gamma+t}$ holds for $t$. Using equation 38, we can write

$$\|\boldsymbol{\psi}_{t+1} - \boldsymbol{\theta}^*\|^2 \leq (1 - \eta_t\mu)\|\boldsymbol{\psi}_t - \boldsymbol{\theta}^*\|^2 + \eta_t^2 B$$
$$\leq \left(1 - \frac{\beta\mu}{t+\gamma}\right)\frac{v}{t+\gamma} + \frac{\beta^2 B}{(t+\gamma)^2}$$
$$= \frac{t+\gamma-1}{(t+\gamma)^2}v + \left[\frac{\beta^2 B}{(t+\gamma)^2} - \frac{\beta\mu-1}{(t+\gamma)^2}v\right] \leq \frac{v}{t+\gamma+1} \tag{49}$$

which proves that $\mathbb{E}_{\mathbb{S}_t}[\|\boldsymbol{\rho}_{t+1} - \boldsymbol{\theta}^*\|^2] \leq \frac{v}{\gamma+t+1}$ holds. Choosing $\beta = \frac{2}{\mu}$ and $\gamma = \max\{\frac{8L}{\mu}, m\} - 1$, we have $\eta_t = \frac{2}{\mu(\gamma+t)}$. It can be verified that in this case $\eta_t \leq 2\eta_{t+m}$. Then, we can write

$$v = \max\left\{\frac{\beta^2 B}{\beta\mu - 1}, (\gamma + 1)\|\boldsymbol{\psi}_1 - \boldsymbol{\theta}^*\|^2\right\} \leq \frac{\beta^2 B}{\beta\mu - 1} + (\gamma + 1)\|\boldsymbol{\psi}_1 - \boldsymbol{\theta}^*\|^2$$
$$\leq \frac{4B}{\mu^2} + (\gamma + 1)\|\boldsymbol{\psi}_1 - \boldsymbol{\theta}^*\|^2. \tag{50}$$

Combining equation 50 with equation 49 and the fact that $\mathbb{E}_{\mathbb{S}_t}[\boldsymbol{\rho}_t] = \boldsymbol{\psi}_t$, we get

$$\mathbb{E}_{\mathbb{S}_t}[\|\boldsymbol{\rho}_{t+1} - \boldsymbol{\theta}^*\|^2] = \|\boldsymbol{\psi}_{t+1} - \boldsymbol{\theta}^*\|^2 \leq \frac{1}{t+\gamma+1}\left(\frac{4B}{\mu^2} + (\gamma + 1)\|\boldsymbol{\psi}_1 - \boldsymbol{\theta}^*\|^2\right) \tag{51}$$

According to smoothness assumption in A 1, we can write

$$\mathbb{E}[\mathcal{L}(\pi_{\boldsymbol{\rho}_t})] - \mathcal{L}(\pi_{\boldsymbol{\theta}^*}) \leq \frac{L}{2}\mathbb{E}_{\mathbb{S}_t}[\|\boldsymbol{\rho}_{t+1} - \boldsymbol{\theta}^*\|^2] \tag{52}$$

Combining equation 52 with equation 51, we arrive at

$$\mathbb{E}[\mathcal{L}(\pi_{\boldsymbol{\rho}_t})] - \mathcal{L}(\pi_{\boldsymbol{\theta}^*}) \leq \frac{L}{2(t+\gamma)}\left(\frac{4B}{\mu^2} + (\gamma + 1)\|\boldsymbol{\psi}_1 - \boldsymbol{\theta}^*\|^2\right). \tag{53}$$

Plugging in $\boldsymbol{\psi}_1 = \boldsymbol{\theta}_{\text{ref}}$ in equation 53 proves the Theorem. Note that for the ease of notation we drop the expectation in Theorem 1. Furthermore, it should be noted that most of proof steps are taken from [33]. Furthrmore, it is useful to mention that Theorem 1 is proved for the case where $w_1 = \ldots = w_N = \frac{1}{N}$. However, extension to non-uniform cases is straightforward. Define the scaled loss for the objective $i$ as $\tilde{\mathcal{L}}_i(\pi_{\boldsymbol{\theta}}) = w_i N\mathcal{L}(\pi_{\boldsymbol{\theta}})$. Then it can be concluded that $\mathcal{L}(\pi_{\boldsymbol{\theta}}) = \frac{1}{N}\sum_{i=1}^{N}\tilde{\mathcal{L}}_i(\pi_{\boldsymbol{\theta}})$. Therefore, in that case the proof can be applied to scaled losses.

## B  Supplementary Experimental Results and Details

This appendix provides supplementary experimental results and implementation details.

### B.1  Implementation Details for Small Molecule Generation

To fine-tune the MolGPT-2 model using MORLHF, RS, and IterativeRS, we employed PPO from the TRL library. For each objective, a reward model was trained using the labeled QM9 dataset. Each reward model consists of a MolGPT-2 backbone with a three-layer MLP head; only the MLP head was trained. The dataset was split into 80% training, 10% validation, and 10% test sets. All models were fine-tuned with a learning rate of $1.41 \times 10^{-5}$ using the Adam optimizer and a batch size of 128. The RiC baseline was configured with the same hyperparameters and settings as MORLHF, RS, and IterativeRS. We set $p = 2$ for RiC. Model training was conducted using four V100 GPUs. To perform merging for IterativeRS and RS, we average all objective-specific model weights. For IterativeRS the number of selected objectives was 3.

## B.2 Implementation Details for DNA Sequence Generation

Similar to the molecule generation task, we fine-tuned the DNAGPT-2 model using MORLHF, RS, and IterativeRS with PPO from the TRL library. A subset of 100,000 samples was uniformly sampled from the MPRA dataset and evaluated using the Malinois model to obtain activity scores across three cell lines. This subset was used as the labeled dataset to train the reward models for PPO. For each objective, a separate reward model was trained using this labeled data. Each reward model consists of a DNAGPT-2 backbone with a three-layer MLP head, where only the MLP head was trained. The dataset was split into 70% training, 10% validation, and 20% test sets. All models were fine-tuned with a learning rate of $1.41 \times 10^{-5}$ using the Adam optimizer and a batch size of 128. The RiC baseline used the same hyperparameters and settings as MORLHF, RS, and IterativeRS. We set $p = 2$ for RiC. Model training was performed on four V100 GPUs. For IterativeRS the number of selected objectives was 3.

## B.3 Implementation Details for Text Summarization

To fine-tune the Llama-3.2-3B-Instruct model using MORLHF, RS, and IterativeRS, we employed PPO from the TRL library. We first passed all prompt–response pairs from the Reddit dataset through three oracle reward models to construct a multi-labeled dataset. For each objective, a proxy reward model was trained using the dataset and the corresponding objective-specific labels. Each proxy model consists of a Llama-3.2-3B-Instruct backbone with a two-layer MLP head, with only the MLP head being trained. We used the training set from the Reddit dataset for training and randomly split its validation set into two subsets to serve as validation and test sets. The validation set was used both for training the proxy reward models and for supervised fine-tuning in RiC. During inference, prompts from the test set were provided to the fine-tuned models to generate text summaries. The maximum summary length was set to 32.

Before applying PPO fine-tuning, we first trained an SFT model. The PPO fine-tuning was then performed using this SFT model. We observed that initializing PPO with an SFT model leads to improved ROUGE scores in the generated summaries. To construct the SFT training dataset, for each prompt in the training set, we selected the summary that outperformed the alternative in the majority of objectives based on reward scores. SFT was performed for two epochs with a learning rate of $1.41 \times 10^{-6}$ using this constructed dataset. For fine-tuning with PPO, we selected a random rollout of 1,024 samples per epoch and used a batch size of 128. Each model was fine-tuned for 20 epochs using a learning rate of $1.41 \times 10^{-6}$ and the Adam optimizer. The RiC baseline was fine-tuned on the entire training set for 2 epochs, using the same learning rate and batch size. We set $p = 2$ for RiC. All other hyperparameters were kept consistent across MORLHF, RS, and IterativeRS. Training was conducted on four A100 GPUs. For IterativeRS, the number of selected objectives was set to 3. For both RS and IterativeRS, model merging was performed by selecting from seven candidate merged models obtained using seven different sets of merging weights. The model with the highest average reward, as evaluated by the reward models, was selected. The seven sets of merging weights consisted of $[1/3, 1/3, 1/3]$, all permutations of $[1/6, 1/6, 2/3]$, and all permutations of $[1/6, 5/12, 5/12]$. During training, at each merging step, we computed the reward of each merged model on 256 samples from the training data and selected the model with the highest average reward for the next iteration. After training, to obtain the final merged model, we evaluated all seven merged models on 1,024 validation samples and selected the one with the highest average reward. We then assessed its performance on the test set. Note that RS does not perform merging during training.

## B.4 Supplementary Results

We performed additional experiments on the DNA sequence generation task to evaluate the performance of RL-based fine-tuning methods MORLHF, RS, and IterativeRS, using RLOO, which does not rely on a value model. The results are presented in Table 5. The results show that IterativeRS achieves a higher average reward than both MORLHF and RS when using RLOO.

To investigate the influence of merging on the performance of IterativeRS, we conducted supplementary experiments. For DNA sequence generation task, we considered ten different sets of merging weights, including $[1/3, 1/3, 1/3]$, permutations of $[1/6, 1/6, 2/3]$, permutations of $[1/6, 5/12, 5/12]$ and permutations of $[1/2, 1/4, 1/4]$. To assess the performance of each merged model, we generated $8,192$ samples per model and identified the Pareto-optimal sequences based on scores from reward

Table 5: Average performance of Pareto-front DNA sequences generated by multi-objective approaches using RLOO.

|  | K562 | HepG2 | SKNSH | Avg Reward | ICV |
|---|---|---|---|---|---|
| MORLHF | 0.3754 | 0.6747 | 0.6882 | 0.5794 | 3.5907 |
| RS | 0.4080 | 0.6571 | 0.6786 | 0.5812 | 5.7214 |
| IterativeRS | 0.3559 | 0.6860 | 0.7061 | 0.5826 | 3.9890 |

Table 6: Comparison of selective merging and fixed merging on the performance of IterativeRS on DNA sequence generation task.

|  | K562 | HepG2 | SKNSH | Avg Reward |
|---|---|---|---|---|
| fixed merging | 0.3559 | 0.6860 | 0.7061 | 0.5826 |
| selective merging | 0.3678 | 0.6778 | 0.6923 | 0.5793 |

models. The final selection was based on the model that achieved the highest average reward score on its Pareto optimal sequences. This method is referred to as *selective merging*, whereas merging with uniform weights is referred to as *fixed merging*. Table 6 presents the results obtained using RLOO. The results suggest that selective merging offers no improvement over fixed merging. One possible reason for the limited effectiveness in the DNA sequence generation task is the evaluation method, which relies on Pareto-optimal samples generated by the model. This makes identifying the best-performing model more challenging. During training, merging allows experts to transfer cross-task knowledge, which can help the final merged model generate higher-quality sequences. However, selecting the best merged model among candidates is difficult because it depends on evaluating the Pareto-optimality of generated sequences using reward models. These reward models are trained on limited data derived from an oracle model used during evaluation, leading to a performance gap between the reward models and the oracle. As a result, assessing Pareto-optimality using these reward models may not yield reliable outcomes.

To examine the effect of the merging strategy on both RS and IterativeRS in the molecule generation task, we applied MolMoE [7] to each method. MolMoE is an expert merging method designed for molecular applications, whereas IterativeRS focuses on expert training. Therefore, these two methods can be used in conjunction. We applied MolMoE to both IterativeRS and RS, and the results are presented in Table 7. As shown, IterativeRS with MolMoE outperforms RS with MolMoE across all objectives. Comparing Table 1 and Table 7, we observe that incorporating MolMoE improves the performance of RS across all objectives. While IterativeRS with MolMoE achieves nearly the same average reward as IterativeRS without MolMoE, it yields an $11\%$ improvement in ICV. It is worth noting that one of the main advantages of using MolMoE is its ability to handle scenarios where preferences over objectives change dynamically over time, which is outside the scope of this paper's experimental study.

## C   Supplementary Related Works

**Federated Learning.** Federated learning involves a group of users, called clients, who collaborate with each other through communication with a central server to train a global model [39, 19, 57]. There is an analogy between federated learning and multi-objective reinforcement learning in the context of foundation model fine-tuning. In federated learning, the clients and the server work together to train a model that performs optimally across all clients' data. However, this can be challenging since the data may be distributed non-i.i.d. among clients, which similar to multi-objective reinforcement learning can lead to conflicting objectives during model training. To address this issue, several personalized federated learning algorithms have been proposed in the literature [48, 15, 11, 32, 38, 8, 58, 16].

**GFlowNet.** GFlowNets, initially proposed by [4], were introduced as a generative reinforcement learning framework designed to effectively handle scenarios with multiple paths leading to a common state. They have been widely applied to biological sequence [23, 28, 17] and molecule design [61, 29, 27] tasks, where their effectiveness has been well documented. In this paper, we focus on policy gradient–based methods such as PPO, due to their computational efficiency for foundation

Table 7: Average performance of Pareto-optimal molecules generated by RS and IterativeRS employing MolMoE for merging.

|  | $\alpha$ **energy** | **gap** | $U_0$ **energy** | **Avg Reward** | **ICV** |
|---|---|---|---|---|---|
| RS+MolMoE | 1.4499 | 0.9715 | 1.5988 | 1.3400 | 4.1120 |
| IterativeRS+MolMoE | 1.5651 | 0.9941 | 1.6420 | 1.4004 | 3.9938 |

model fine-tuning. It is also worth noting that several methods have been proposed in the literature to improve the learning efficiency of GFlowNets [5, 37, 36, 47]. Furthermore, GFlowNets have recently been utilized to enhance the reasoning capabilities of large language models and vision-language models [25, 55, 54].

## D   Societal Impact

In this paper, we addressed the problem of fine-tuning large language models (LLMs) on multiple objectives—a challenge with significant implications in areas such as small molecule design for drug discovery and biological sequence design. Methods that enable LLMs to generate molecules or biological sequences with desirable functionalities hold great promise for accelerating the discovery of new drugs and therapeutics, potentially benefiting society at large. However, we recognize the dual-use nature of this research. There is also the risk that such technologies could be misused or exacerbate existing health disparities, particularly among marginalized communities. As researchers, we underscore the importance of carefully considering both the societal benefits and the potential unintended consequences of this work. We remain optimistic that the broader impact of our contributions will lean toward positive, equitable outcomes.

