# OpenReview forum: "Iterative Foundation Model Fine-Tuning on Multiple Rewards"
_NeurIPS.cc/2025/Conference — NeurIPS 2025 poster_

### Official Review · Reviewer_2pTb · 2025-06-24

**Clarity:** 3
**Significance:** 2
**Originality:** 3
**Rating:** 4
**Confidence:** 3

**Summary:**

This paper introduces a framework for fine-tuning foundation models by leveraging multiple given rewards functions. The proposed approach unifies two existing techniques, Reward Combining and Reward Soups. Furthermore, the authors provide theoretical support for their method, including an upper bound on the error between the optimal policy and the policy trained using their framework.

**Questions:**

Below are some questions and suggestions for the authors. Addressing these could help clarify certain aspects of the work and potentially strengthen the paper:

1) The proof of Theorem 1 appears to utilize notation that differs from what is used in the main text. This inconsistency can create confusion for the reader. Would the authors consider revising the proof to maintain consistent notation with the rest of the paper? Also, clarity of Theorem 1 proof and Lemma 1: The proof of Theorem 1, especially the transition in notation around lines 432-435 and its connection to Lemma 1, could be made more accessible. The current presentation has some notational overlaps that harm comprehension. To improve clarity and consistency, please consider: 1.1) Adding a table of notations at the beginning of the appendix. 1.2) Providing a detailed proof of Lemma 1 using the same consistent notation as presented in the main text. 1.3) Breaking down the proof of Theorem 1 into more clearly delineated logical steps.

2) The experiments and analysis seem to exclusively use weights $w_i = \frac{1}{N}$. The paper mentions that other settings for these importance weights are possible. Have the authors investigated alternative parametrizations for $w_i$? Presenting results from such explorations could demonstrate the sensitivity of the method to this choice or some different behaviour.

3) Have the authors considered comparing diversity-specific metrics (e.g., entropy, or other established measures) between their method and baselines? In addition, authors mention some GFlowNets baselines [1]. GFlowNet-based approaches are also good at generating more diverse candidates, have you considered a comparison with GFlowNets?

4) In conclusion authors write: “We showed that both MORLHF and Rewarded Soups can be viewed as special cases of IterativeRS, highlighting its greater flexibility for performance enhancement.” Though, in my humble opinion, it is a little hidden, I would rather make a detailed section pinpointing this relation. Although, it is a minor issue of mine.

References:

[1] Jain et al. Multi-Objective GFlowNets. ICML 2023.

**Ethical Concerns:**

["NO or VERY MINOR ethics concerns only"]

**Final Justification:**

I believe that the main issues I pointed out were adequately addressed in the rebuttal: 1) poor performance on the text summarization task, 2) concern in the novelty because of the method being a combination of previous approaches. As they are resolved, I decided to raise my score to borderline accept. I also find the authors rationale about not comparing to GFlowNet-based methods to be reasonable.

**Limitations:**

The paper does not include a dedicated section discussing the potential limitations of the proposed algorithm. Here are my thoughts:

1) The current experimental results do not conclusively demonstrate that the proposed method offers a significant advantage over standard fine-tuning baselines, particularly for tasks outside the biological domain. Further evidence would be needed to establish its superiority in a broader range of applications.

2) While the paper unifies existing techniques, the extent to which this unification yields fundamentally novel insights beyond the combination itself is not fully clarified.

**Paper Formatting Concerns:**

None.

**Quality:**

2

**Strengths And Weaknesses:**

**Quality**:
The paper appears to be technically sound in its formulation. However, the claims made could be more robustly supported by the experimental results.

Strengths: The theoretical derivation of an error bound is a positive aspect. The core idea of unifying Reward Combining and Reward Soups is presented.

Weaknesses: The experimental validation seems limited. There is a need for more extensive experiments to demonstrate the stability and generalizability of the proposed method. For instance, the method shows promise primarily on biological tasks. The results on the text summarization task, however, indicate a marginal improvement only in average reward, while falling behind in other metrics, which is not really representative of the method’s utility. The work could benefit from a more thorough empirical evaluation.

**Clarity**: The paper is generally well-organized, with a coherent structure and logical flow between sections.

Strengths: The overall organization is good.

Weaknesses: At times, the exposition could be clearer, making it somewhat challenging to fully grasp the rationale behind certain choices or implications of some arguments. (See Questions 1 and 4)

**Significance**:
The potential impact of this work on the community is an area of concern.

Strengths: The paper attempts to unify existing techniques, which can be valuable. The inclusion of theoretical analysis is commendable.

Weaknesses: The proposed method appears to be a relatively straightforward combination of two prior techniques. For a method that builds upon existing ideas, especially if the combination seems intuitive, a comprehensive suite of experiments demonstrating clear advantages across diverse tasks would be expected. The current set of experiments (two biological tasks and one text-based task where performance is not compelling) does not fully meet this expectation.

**Originality**:
The work does present a novel combination of existing methods.

Strengths: To the best of my knowledge, the specific unified framework presented has not been proposed previously.

Weaknesses: While the combination is new, the novelty would be more impactful if accompanied by surprising empirical results or deeper, unexpected theoretical insights.

---

> ### Author Rebuttal · Authors · 2025-07-30
>
> Thank you for taking the time to review our paper and sharing your valuable thoughts and suggestions. Please find our responses to your comments below.
>
> We would like to clarify that the proposed algorithm is **not** a combination of MORLHF and RS.  The paper provides a new perspective that MORLHF can be viewed as the case where merging is done after every batch. However, if merging happens after multiple batches, the algorithm after each merging is no longer equivalent to MORLHF.
>
> ## Experiments
> We performed further experiments on the text summarization task. To improve the performance of the proposed algorithm, we considered seven different sets of merging weights, including [1/3, 1/3, 1/3], permutations of [1/6, 1/6, 2/3], and permutations of [1/6, 5/12, 5/12]. During training, at each merging step, we calculated the reward of each merged model on the training data and chose the one with the highest average reward for the next iteration. To obtain the final merged model after training, we measured the average reward of all seven merged models and chose the one with the highest average reward on the validation data. We then tested the performance on the test set. We followed the same procedure for RS as well. Note that RS does not perform merging during training. The table below shows the new results. These results demonstrate the effectiveness of the flexible merging strategy in IterativeRS, as it significantly outperforms other methods on all metrics.
> |  | faithful  | summary | deberta | Avg Reward | Sharp-like |
> | :--------- | --------: | :---------: | :--------: | :------------: | :----------: |
> | RS | 0.6549 | 0.4644 | 0.3178 | 0.4791 | 3.3719 |
> | IterativeRS | 0.7023 | 0.5160 | 0.5834 | 0.6006 | 6.2180 |
>
> ### Diversity and GFlowNet
> The focus of this paper is on reinforcement learning for LLM fine-tuning. Due to pre-training on vast amounts of unlabeled data, LLMs are known to generate diverse outputs. During fine-tuning, we monitor the KL divergence between the policy and a reference policy to ensure that the fine-tuned model does not deviate too far from the initial pre-trained model. For example, we set kl_coef = 0.05 during PPO fine-tuning.
>
> Sequence generation with GFlowNet can be viewed as a soft Q-learning problem [15]. However, applying Q-learning to LLM fine-tuning may not be computationally feasible. Therefore, we did not consider employing or comparing against existing GFlowNet methods. Note that in all experiments, we employ LLMs for sequence generation.
>
> ## Notation and Presentation:
> Since Lemma 1 is taken from [19], we wanted to retain its original notation. We understand the confusion this may have caused. To address this, we will unify the notation of Lemma 1 with the rest of the paper and add its proof to the appendix. Furthermore, based on your suggestion, we will include a notation table at the beginning of the appendix.
>
> Please let us know if you have further comments and concerns.

---

> > ### Comment · Reviewer_2pTb · 2025-08-05
> >
> > I appreciate the authors detailed rebuttal and their efforts in addressing the weaknesses outlined in my review.
> >
> > Firstly, I recommend incorporating the clarifications regarding MORLHF and RS into the paper in the form of a more clear separate discussion. Upon re-reading, I still find it difficult to fully understand this concept based solely on the text.
> >
> > Regarding my comments about GFlowNets, while there are studies that apply them to RL fine-tuning related tasks for large language models [1], I find the authors rationale about not comparing to GFlowNet-based methods to be reasonable.
> >
> > Considering the additional experimental results with improved metrics and the clarifications provided by the authors, I will raise my score to borderline accept.
> >
> > References:\
> > [1] Hu et al. Amortising intractable inference in large language models. ICLR 2024

---

> > > ### Author Response · Authors · 2025-08-06
> > >
> > > Thank you for your feedback on our rebuttal.
> > >
> > > To better emphasize the differences between MORLHF and RS, we will add comments in the paper regarding the relationship between merging and MORLHF. Specifically, we will clarify that our perspective frames MORLHF as an algorithm that involves merging.
> > >
> > > We will also expand the discussion of GFlowNet in our related works section by including studies that connect GFlowNet to LLMs. However, for our experiments, we chose policy gradient–based methods due to their computational efficiency for LLM fine-tuning. Thank you again for your review and comments.

---

### Official Review · Reviewer_WYNS · 2025-06-26

**Clarity:** 3
**Significance:** 2
**Originality:** 3
**Rating:** 4
**Confidence:** 3

**Summary:**

This paper addresses the fine-tuning problem of foundation models in multi-reward scenarios and proposes the IterativeRS algorithm. By iteratively fine-tuning expert policies for each objective and periodically merging parameters, it balances objective-specific skills and policy consistency. Theoretical analysis proves its convergence superiority over baselines such as MORLHF and Rewarded Soups under convex loss assumptions, providing a more flexible and effective solution for multi-objective reinforcement learning fine-tuning.

**Questions:**

Please refer to the above weakness.

**Ethical Concerns:**

["NO or VERY MINOR ethics concerns only"]

**Final Justification:**

1. The theory focuses on justifying the algorithm' s design is logically consistent.
2. The subset strategy raises critical issues: vague selection rules, added hyperparameter complexity, and reduced generality for full-objective optimization. Not accepted. We recommend either clarifying subset selection rules or explicitly discussing their limitations.
3. Expanded experiments strengthen empirical support.
4. The rationale for excluding GFlowNet (computational infeasibility for LLM fine-tuning) is valid. Adding MoMoE comparisons improves baseline coverage.
In view of the explanations from authors, I will increase my score accordingly.

**Limitations:**

1. Theoretical analysis relies on assumptions like convexity and smoothness of loss functions (e.g., A1–A3), but loss functions in practical multi-objective optimization are often non-convex, possibly causing deviations between theoretical performance bounds and actual performance when objectives conflict.

2. The mechanism of iteratively training expert policies and merging parameters leads to superlinear growth of computational cost with the number of objectives N and merging frequency m. For example, in DNA sequence generation experiments, the training time is about 3 times that of MORLHF, restricting large-scale applications.

3. Tasks such as text summarization do not analyze the impact of weight configurations on results by combining domain-standard metrics (e.g., ROUGE scores), and lack targeted discussions on practical scenarios (e.g., news summarization).

4. Experiments only compare methods like MORLHF and Rewarded Soups, without including emerging multi-objective optimization algorithms (e.g., Multi-Objective GFlowNets), and lack comparison with professional methods (e.g., Mol-MOE) in biological sequence generation, affecting the universality of conclusions.

**Quality:**

2

**Strengths And Weaknesses:**

Strength:
1. The paper proposes a novel IterativeRS method. Through iterative training and merging of expert policies, it provides a unique alternative to existing methods such as MORLHF, and addresses multi - objective optimization problems in a more principled manner.
2. It conducts a comprehensive theoretical analysis of the convergence behavior of IterativeRS, which not only explains the characteristics of this method but also provides a theoretical basis for understanding related baseline methods.
3. Extensive experiments are carried out in multiple tasks such as small - molecule generation, DNA sequence generation, and text summarization. By comparing with a variety of baseline methods, the advantages of IterativeRS in terms of average reward and cross - objective consistency are verified.
Weakness
1. The theoretical analysis relies on assumptions such as the convexity of the loss function, which may not hold in real - world scenarios. This leads to limited universality of the theoretical results, and the performance of the method in complex practical applications may deviate from theoretical predictions.
2. The method involves iterative merging and multi - objective optimization, resulting in a relatively high computational complexity. Although the trade-offs of hyper-parameter tuning are discussed, it may still face challenges in terms of computational resources under complex configurations, restricting its practical application and promotion.
3. Although the method is applied to the text summarization task, there is a lack of in - depth interpretation of the actual application scenarios of the generated summaries. It fails to analyze the specific impact of different reward model weights on the quality of summaries in combination with domain - specific indicators such as faithfulness and readability.
4. The number of baseline methods used in the experiments is relatively small. Incorporating more baseline methods widely adopted in relevant sub - fields could lead to a more rigorous and comprehensive evaluation of IterativeRS.

---

> ### Author Rebuttal · Authors · 2025-07-30
>
> We would like to express our gratitude for your valuable comments and suggestions. Please find our responses to your comments below.
> ## Theoretical Analysis
> The conclusions drawn from the theoretical analysis in the paper are not guaranteed to hold for non-convex deep neural networks, which are typically used in practice. However, the goal of the theoretical analysis is not to fully explain the behavior of the proposed algorithm in practical scenarios. Instead, it demonstrates that the algorithm’s design is both sensible and grounded in a rigorous theoretical foundation. Finally, the theoretical convergence analysis of deep neural networks used in practice remains an open research problem and is beyond the scope of this paper. It is also useful to mention that assumptions A1–A3 are common in convex optimization.
> ## Complexity
> The worst-case computational and memory complexities of IterativeRS are of the same order as those of RS. Both algorithms update the model over $T$ batches. IterativeRS performs merging every $m$ rounds; however, the cost of merging is negligible compared to expert training, as it typically does not involve additional training. Furthermore, as described in line 7 of Algorithm 1, between two merging rounds, a subset of objectives can be selected for fine-tuning. This can significantly reduce the complexity of IterativeRS, especially when compared to RS. Setting $M=1$ results in a complexity comparable to that of MORLHF.
> ## Experiments
> We performed further experiments on the text summarization task. To improve the performance of the proposed algorithm, we considered seven different sets of merging weights, including [1/3, 1/3, 1/3], permutations of [1/6, 1/6, 2/3], and permutations of [1/6, 5/12, 5/12]. Previously, we only considered the uniform merging weights [1/3, 1/3, 1/3]. During training, at each merging step, we calculated the reward of each merged model on the training data and chose the one with the highest average reward for the next iteration. To obtain the final merged model after training, we measured the average reward of all seven merged models and chose the one with the highest average reward on the validation data. We then tested the performance on the test set. We followed the same procedure for RS as well. Note that RS does not perform merging during training. The table below shows the new results. These results demonstrate the effectiveness of the flexible merging strategy in IterativeRS, as it significantly outperforms other methods on all metrics.
>
> |  | faithful  | summary | deberta | Avg Reward | Sharp-like |
> | :--------- | --------: | :---------: | :--------: | :------------: | :----------: |
> | RS | 0.6549 | 0.4644 | 0.3178 | 0.4791 | 3.3719 |
> | IterativeRS | 0.7023 | 0.5160 | 0.5834 | 0.6006 | 6.2180 |
>
> In addition, to incorporate more standard metrics into our evaluation, we define a new metric as the average of all rewards and the ROUGE score. The results are presented in the table below. We can either replace the "Avg Reward" column in Table 3 with this new metric or update the definition of Avg Reward accordingly. As can be seen, IterativeRS achieves the highest average score.
>
> | methods | MORLHF  | RS | RiC | IterativeRS |
> | :--------- | --------: | :---------: | :--------: | :------------: |
> | Avg Score | 0.3943 | 0.3925 | 0.4518 | 0.4785 |
>
> ### Baselines
> To the best of our knowledge, we have compared the proposed algorithm with all relevant state-of-the-art multi-objective reinforcement learning methods for LLM fine-tuning. Sequence generation with GFlowNet can be viewed as a soft Q-learning problem [15]. However, applying Q-learning to LLM fine-tuning may not be computationally feasible. Therefore, we did not consider employing or comparing against existing GFlowNet methods. Note that in all experiments, we employ LLMs for sequence generation.
>
> IterativeRS and MolMoE have distinct and non-overlapping functionalities and applications. MolMoE is an expert merging method designed for molecular applications, whereas IterativeRS focuses on expert training. Therefore, these two methods can be used in conjunction. We applied MolMoE to both IterativeRS and RS, and the results are presented below. One of the main advantages of using MolMoE is its ability to handle scenarios where preferences over objectives change dynamically over time, which is outside the scope of this paper's experimental study.
>
> |  | $\alpha$ energy | gap | $U_0$ energy | Avg Reward | Sharp-like |
> | :--------- | --------: | :---------: | :--------: | :------------: | :----------: |
> | RS+MolMoE | 1.4499 | 0.9715 | 1.5988 | 1.3400 | 4.1120 |
> | IterativeRS+MolMoE | 1.5651 | 0.9941 | 1.6420 | 1.4004 | 3.9938 |
>
> Please let us know if you have further comments and concerns.

---

> > ### Comment · Reviewer_WYNS · 2025-08-07
> > **I will increase my score**
> >
> > 1. The theory focuses on justifying the algorithm' s design is logically consistent.
> > 2. The subset strategy raises critical issues: vague selection rules, added hyperparameter complexity, and reduced generality for full-objective optimization. Not accepted. We recommend either clarifying subset selection rules or explicitly discussing their limitations.
> > 3. Expanded experiments strengthen empirical support.
> > 4. The rationale for excluding GFlowNet (computational infeasibility for LLM fine-tuning) is valid. Adding MoMoE comparisons improves baseline coverage.
> > In view of the explanations from authors, I will increase my score accordingly.

---

> > > ### Author Response · Authors · 2025-08-08
> > >
> > > Thank you for your feedback on our rebuttal. Regarding the objective subset selection, the selection rule is uniform at random. As stated in line 149 on page 4: "*To reduce computational complexity, a subset of $M \le N$ objectives can be selected uniformly at random at each merging step...*". We can clarify this further by explicitly including the mathematical expression for the uniform sampling of $M \le N$ objectives. Thank you again for your review and comments.

---

### Official Review · Reviewer_ysgq · 2025-07-02

**Clarity:** 4
**Significance:** 2
**Originality:** 2
**Rating:** 3
**Confidence:** 4

**Summary:**

In this paper, the authors introduce an algorithm for performing fine-tuning on foundation models with multi-objective rewards. In particular, they propose a generalization of current methods which either (1) train separate models on each objective and merge parameters at the end or (2) train a single model on all parameters with a weighted reward function. The generalization simply performs an iterative merging procedure where separate models are trained in parallel for $m$ steps and then merged before continuing another round of independent training. They derive a theoretical bound on the performance gap in the training loss between their proposed method and the optimal policy for the combined reward function and offer some insights on hyperparameter selection based on this bound. Finally, they provide experimental justification for their method on three practical and pertinent fine-tuning domains, demonstrating out-performance of their method in terms of average reward across all domains.

**Questions:**

Questions:

(1 How do you compute the Pareto front? Are the results in Table 1-3 only for generated results on the Pareto front? Wouldn’t it be necessary to report how many results generated were on this front for each algorithm?

(2) What exactly is the Sharp-like score? Is it better to be high or low? How should we balance the Sharp-like score versus the average reward?

(3) How can we interpret Figs 1-3. e.g. in Line 265 “Notably, Figure 1 shows that the highest-scoring molecules are produced by IterativeRS” – From the plots, it is hard to varify this claim from the plots – the Iterative RS and MORLHF distributions seem to be mostly overlapping. This is true in most of the Fig 1-3 results.

**Ethical Concerns:**

["NO or VERY MINOR ethics concerns only"]

**Final Justification:**

The original results in the paper do not provide sufficient performance improvement over prior methods to be convincing. While the authors provided some additional results during the rebuttal period on the text domain that show stronger outperformance, they did not show similar outperformance in the other domains.

**Limitations:**

The authors do not discuss any limitations of their work. They do note a limitation of the theoretical analysis, that it does not apply to generalization results, but this should probably be restated in the conclusion. Other limitations may include the reliance on knowing the $w$ weights to balance the value of various rewards, which is impractical in most scenarios.

**Quality:**

2

**Strengths And Weaknesses:**

Strengths:
(1) The paper is very clearly written and easy to follow.
(2) The experiments cover three diverse use cases and hence show a broad range of practical application domains for the proposed algorithm. The practical domains examined are relevant and important domains for the research community.
(3) The authors provide both theoretical and experimental results for their proposed algorithm

Weaknesses:
(1) The results are not wholly convincing. From Figs 1-3, the only clear out-performance is versus RiC and it is somewhat unclear why this is a relevant baseline. It would be more interesting to see out-performance versus MORLHF and RS since, as the authors note, these are two extremes of the proposed IterativeRS algorithm. While the results in Tables 1-3 show that IterativeRS consistently outperforms in terms of average reward on the Pareto front, it is not exactly clear how to interpret this result (see questions below). Moreover, it is not clear if the out-performance is statistically significant since no confidence intervals are reported. I am also not clear on how to weigh the average reward results versus the Sharp-like score (and it is not clear what the Sharp-like score is measuring).I would be willing to increase my score if these issues are resolved.

(2) The utility of the insights from the theoretical analysis seem limited. The theoretical results do not indicate any theoretical justification for IterativeRS over MORLHF or RS. The authors claim that the theoretical results provide insights into the behavior of these various algorithms, however, this claim is not clearly demonstrated. The authors themselves note that the results are limited to the training loss and do not apply to generalization capabilities which is of most interest. Moreover, the analysis of the hyperparameter choices, particularly the choice of $m$ do not seem to offer much practical use for selecting the hyperparameters. How was $m$ chosen in each of the three experimental cases? Was the theory provided leveraged to provide any insight into the choice of $m$ or was a hyperparameter sweep conducted?

---

> ### Author Rebuttal · Authors · 2025-07-31
>
> Thank you so much for reviewing our paper and letting us know your valuable comments. Please find below our responses to your comments.
>
> ## Theory
> The theoretical analysis in this paper focuses on convex settings. The statement regarding generalization in practice refers to extending the results to non-convex cases, which is the relevant scenario for LLM fine-tuning. The theoretical convergence analysis of deep neural networks remains an open research problem and is beyond the scope of this paper. The purpose of our theoretical analysis is not to fully explain the algorithm’s behavior in practical settings, but rather to show that the algorithm’s design is well-motivated and grounded in a rigorous theoretical foundation.
>
> Furthermore, the goal of this paper is not to propose an algorithm whose superiority over RS and MORLHF is theoretically proven. Instead, the paper suggests that better empirical results than both RS and MORLHF can be achieved by appropriately tuning the hyperparameter $m$ of the proposed IterativeRS.
>
> ## Metrics and Interpretation of Results
> **Sharp-like score:** For a given sample, the Sharpe-like score is defined as the average reward across all objectives divided by the standard deviation of those rewards. A higher Sharpe-like score indicates lower variability and more balanced performance across objectives. Multiplying the reward vector by a scalar $\lambda>1$ increases the standard deviation, but the Sharpe-like score remains unchanged. In this case, the relative performance across objectives does not change, even though all objective values improve. A higher Sharpe-like score reflects more consistent performance across objectives, which is generally desirable. However, in practice, higher average performance (i.e., average reward) may be preferred over higher consistency. As an example, if two objectives remain unchanged while one objective improves, this leads to a reduction in the Sharpe-like score, while the overall performance has improved. The definition of the Sharpe-like score is provided in the first paragraph of Section 5. We can highlight the definition by providing math formulation.
>
> **Pareto-front:** Let us first explain the difference between the goals in text summarization and biological applications. In text summarization, each input consists of a prompt, and the goal is to generate a high-quality summary for that prompt. In contrast, for small molecule and DNA sequence generation, there is no prompt; the objective is to discover a small number of high-performing sequences using the LLM. In our experiments, we used the fine-tuned model to generate 10,000 molecules or DNA sequences, which were then evaluated using oracle models. Note that in our setup, oracles are distinct from reward models. Reward models are trained on labeled training data, where the labels are provided by oracle models. In biological applications, if one generated molecule outperforms another (both produced by the same model) across all objectives, the former dominates the latter and is included in the Pareto front, while the dominated sample is considered suboptimal. Therefore, for molecule and DNA sequence generation, we report the performance based on the Pareto front. In contrast, for text summarization, we generate a response for each prompt in the test set and report the scores assigned by the oracle reward models. As with the biological tasks, the LLMs did not interact with the oracle models during fine-tuning for text summarization.
>
> **Figures:** The comparison between RiC and IterativeRS is emphasized in the figures because RS and MORLHF can be viewed as special cases of IterativeRS. In the results shown in Figure 1, it is desirable to have samples in the top-right corner. Since the task is molecule generation, generating a few dominant samples is considered optimal. As shown in Figure 1, IterativeRS achieves the highest density of samples in the top-right corner.
>
> ## Experiments
> We performed further experiments on the text summarization task. To avoid relying on predefined merging weights, we considered seven different sets of merging weights, including [1/3, 1/3, 1/3], permutations of [1/6, 1/6, 2/3], and permutations of [1/6, 5/12, 5/12]. Previously, we only considered the uniform merging weights [1/3, 1/3, 1/3]. During training, at each merging step, we calculated the reward of each merged model on the training data and chose the one with the highest average reward for the next iteration. To obtain the final merged model after training, we measured the average reward of all seven merged models and chose the one with the highest average reward on the validation data. We then tested the performance on the test set. We followed the same procedure for RS as well. Note that RS does not perform merging during training. The table below shows the new results. These results demonstrate the effectiveness of the flexible merging strategy in IterativeRS, as it significantly outperforms other methods on all metrics.
>
> |  | faithful  | summary | deberta | Avg Reward | Sharp-like |
> | :--------- | --------: | :---------: | :--------: | :------------: | :----------: |
> | RS | 0.6549 | 0.4644 | 0.3178 | 0.4791 | 3.3719 |
> | IterativeRS | 0.7023 | 0.5160 | 0.5834 | 0.6006 | 6.2180 |
>
> **Hyperparameter Tuning:** To tune the hyperparameter $m$, we perform a few experiments with different values of $m$ and evaluate performance on the validation set. Based on these results, we select the value of $m$.
>
> Please let us know if you have further comments and concerns.

---

> > ### Comment · Reviewer_ysgq · 2025-08-04
> >
> > Thank you for your response.
> >
> > Based on your responses and having read the other reviewers comments, I still think the experimental results are not wholly convincing. Since the main contribution of the paper is experimental (the algorithm is a simple combination of existing methods and the theory has limited utility), I think more significant outperformance would need to be demonstrated. I appreciate the inclusion of the new results with variable weights during merging. These results are more convincing than the fixed weights in the paper - did you repeat these experiments in the other domains?

---

> > > ### Author Response · Authors · 2025-08-04
> > >
> > > Thank you for your feedback. We would like to clarify that the proposed algorithm is **not** a simple combination of existing methods. Instead, it introduces a generalized approach. While incorporating a variable merging weight strategy can further enhance results in tasks such as small molecule and DNA sequence generation, our current results show that IterativeRS already outperforms other baselines, including MORLHF and RS. This highlights the effectiveness of the generalized IterativeRS algorithm.

---

> > > > ### Author Response · Authors · 2025-08-08
> > > >
> > > > We followed up on our discussion regarding the impact of variable merging on results in other domains by applying this approach to DNA sequence generation. However, we found that variable merging was not effective for this domain. Below are the results when using the RLOO fine-tuning framework. We considered ten different sets of merging weights, including [1/3, 1/3, 1/3], permutations of [1/6, 1/6, 2/3], permutations of [1/6, 5/12, 5/12] and permutations of [1/2, 1/4, 1/4]. To assess the performance of each merged model, we generated 8,192 samples per model and identified the Pareto-optimal sequences based on scores from reward models. The final selection was based on the model that achieved the highest average reward score on its pareto optimal sequences.
> > > >
> > > > |  | K562 | HepG2 | SKNSH | Avg Reward |
> > > > | :--- |:---: | :---: | :---: | :----: |
> > > > | fixed merging | 0.3559 | 0.6860 | 0.7061 | 0.5826 |
> > > > | variable merging | 0.3678 | 0.6778 | 0.6923 | 0.5793 |
> > > >
> > > > One possible reason for the limited effectiveness in the DNA sequence generation task is the evaluation method, which relies on Pareto-optimal samples generated by the model. This makes identifying the best-performing model more challenging. During training, merging allows experts to transfer cross-task knowledge, which can help the final merged model generate higher-quality sequences. However, selecting the best merged model among candidates is difficult because it depends on evaluating the Pareto-optimality of generated sequences using reward models. These reward models are trained on limited data derived from an oracle model used during evaluation, leading to a performance gap between the reward models and the oracle. As a result, assessing Pareto-optimality using these reward models may not yield reliable outcomes. Also it is worth noting that the results in the paper show that the IterativeRS can achieve better performance than RS and MORLHF in DNA sequence generation. We can add this new results and discussion to the paper.
> > > >
> > > > Thank you again for your review and comments.

---

### Official Review · Reviewer_Bpig · 2025-07-02

**Clarity:** 3
**Significance:** 2
**Originality:** 3
**Rating:** 4
**Confidence:** 3

**Summary:**

This paper proposes a new multi-objective reinforcement learning method (IterativeRS) that generalizes both multi-objective RLHF (MORLHF) and Rewarded Soups. Existing MORLHF approaches typically optimize a single expert using a scalar-weighted sum of multiple objectives. This approach lacks expert diversity (since there’s only one expert) and is prone to collapsing toward the objective with the dominant reward signal—a well-known challenge in balancing multiple rewards. In contrast, the rewarded soups method trains separate experts for each objective and mixes their parameters post-training. While this allows for expert diversity, it can result in high variance and conflict between experts due to a lack of synchronization. The proposed method introduces **periodic synchronization** using the hyperparameter *m*, effectively balancing objective-level and expert-level variance. By tuning *m*, the method achieves stable performance across three tasks: QM9 molecular generation, DNA sequence design, and Reddit summarization.

**Questions:**

- This method appears to be a general multi-objective RL approach, not limited to fine-tuning. Have you tried applying it to RL training from scratch?
- In the context of LLM fine-tuning—currently the most common use case for RLHF—what are realistic scenarios where multiple objectives are necessary? Can this approach be applied effectively to tasks like text summarization?

**Ethical Concerns:**

["NO or VERY MINOR ethics concerns only"]

**Final Justification:**

I have read the authors' rebuttal. They addressed my concerns related to generalizability, but still, my concerns about scalability are not fully resolved, since the experiments are conducted on short sequence generation tasks. I'll keep my score 4.

**Limitations:**

there is no section for the limitations, please include these.

**Quality:**

3

**Strengths And Weaknesses:**

**Strengths:**

- The paper proposes an elegant algorithm that unifies two existing approaches to multi-objective RL: MORLHF and rewarded soups.
- It highlights the importance of balancing trade-offs between expert-level and objective-level variance.
- The work provides both theoretical insight and empirical validation.

**Weaknesses:**

- The method is only evaluated using PPO. It is unclear how well it generalizes to other RL methods, such as value-based approaches.
- The scale of the applications is relatively small. It is questionable whether this method is necessary for fine-tuning foundation models, as the tasks could potentially be handled by training from scratch.

---

> ### Author Rebuttal · Authors · 2025-07-31
>
> We would like to express our gratitude for reviewing our paper and letting us know your valuable comments. Please find below our responses to your comments.
>
> ## Fine-Tuning Frameworks
> The experiments are conducted using PPO, a commonly used reinforcement learning (RL) method for LLM fine-tuning. Based on your comment, we performed additional experiments on the DNA sequence generation task to evaluate the performance of RL-based fine-tuning methods MORLHF, RS, and IterativeRS, using RLOO, which does not rely on a value model. The experimental setup for RLOO is the same as that used with PPO. The results are presented in the table below. They indicate that using PPO leads to better performance for IterativeRS. It is possible that improved tuning of the hyperparameter $m$ could enhance the performance of IterativeRS when using RLOO. Furthermore, the results show that IterativeRS achieves a higher average reward than both MORLHF and RS when using RLOO.
>
> |  | K562 | HepG2 | SKNSH | Avg Reward | Sharp-like |
> | :----: | :----: | :----: | :----: | :----: | :----: |
> | MORLHF | 0.3754 | 0.6747 | 0.6882 | 0.5794 | 3.5907|
> | RS | 0.4080 | 0.6571 | 0.6786 | 0.5812 | 5.7214 |
> | IterativeRS | 0.3559 | 0.6860 | 0.7061 | 0.5826 | 3.9890|
>
> ## Reinforcement Learning from Scratch
> The focus of this paper is on RL for LLM fine-tuning. However, the proposed method can also be applied to RL from scratch. It is worth noting that even in relatively small-scale applications where training from scratch is feasible, fine-tuning LLMs can offer significant advantages. First, LLMs can be pre-trained on both unsupervised and supervised data, while RL from scratch utilizes labeled data. This pre-training provides LLMs with prior knowledge that can be beneficial for downstream tasks. Moreover, fine-tuning an LLM might be more computationally efficient than training an RL model from scratch. For these reasons, we chose to focus on LLM fine-tuning in this work.
>
> ## Applications
> There are many practical applications for multi-objective RL in LLM fine-tuning. For example, LLMs are currently used for small molecule generation in drug discovery. These models learn to generate valid SMILES representations of molecules through pre-training on large molecular datasets. During generation, the goal is to produce molecules that satisfy multiple desirable properties. Each property typically has its own labeled dataset. An effective fine-tuning approach should enable the LLM to optimize all target properties simultaneously. Similarly, in text summarization, multiple criteria such as helpfulness, conciseness, and faithfulness are important for evaluating the quality of a summary. Relying on a single reward model to capture all these criteria may be suboptimal. Therefore, multi-objective RL offers a necessary solution for a range of important tasks in LLM fine-tuning.

---

> > ### Comment · Reviewer_Bpig · 2025-08-04
> >
> > Thanks for the rebuttal. I'll keep my score 4.

---

> > > ### Author Response · Authors · 2025-08-04
> > >
> > > Thank you again for your review and feedback.

---

### Official Review · Reviewer_e3du · 2025-07-04

**Clarity:** 3
**Significance:** 2
**Originality:** 3
**Rating:** 4
**Confidence:** 2

**Summary:**

The paper proposed a multi-objective reinforcement learning algorithm called IterativeRS to fine-tune foundation models on multiple reward signals. Specifically, the algorithm fine-tunes a separate "expert" policy for each objective to learn objective-specific skills , and then mitigates divergence between these experts by iteratively merging their parameters into a single shared policy, which is then used as the starting point for the next round of fine-tuning. The experiments on diverse tasks including small molecule generation, DNA sequence generation, and text summarization show that IterativeRS generally achieves higher average rewards.

**Questions:**

See weaknesses.

**Ethical Concerns:**

["NO or VERY MINOR ethics concerns only"]

**Final Justification:**

The authors seem to have persistent confusion regarding bounded variance and  bounded random variables. But since theory is not the main contribution, I maintain my stance on weak acceptance.

**Limitations:**

yes

**Quality:**

2

**Strengths And Weaknesses:**

**Strength**

The proposed method generalizes existing methods into a single flexible framework that offers greater potential for performance improvement by controlling the trade-off between specialization and knowledge sharing.

**Weaknesses**

1. Marginal performance gains in text summarization. IterativeRS barely outperforms MORLHF on average reward and is outperformed by RS on the "faithful score" metric. This suggests that the proposed method might have limited applications.
2. The assumption A.3 is too strong and not commonly used in previous literature. The authors should provide a convincing reason on why this is realistic or necessary.
3. Following 2, the theoritical analysis might not be helpful for explaining the practicality of the proposed method.

---

> ### Author Rebuttal · Authors · 2025-07-31
>
> Thank you for reviewing our paper and letting us know your valuable comments. Please find below our responses.
>
> ## Experiments
> We performed further experiments on the text summarization task. To improve the performance of the proposed algorithm, we considered seven different sets of merging weights, including [1/3, 1/3, 1/3], permutations of [1/6, 1/6, 2/3], and permutations of [1/6, 5/12, 5/12]. Previously, we only considered the uniform merging weights [1/3, 1/3, 1/3]. During training, at each merging step, we calculated the reward of each merged model on the training data and chose the one with the highest average reward for the next iteration. To obtain the final merged model after training, we measured the average reward of all seven merged models and chose the one with the highest average reward on the validation data. We then tested the performance on the test set. We followed the same procedure for RS as well. Note that RS does not perform merging during training. The table below shows the new results. These results demonstrate the effectiveness of the flexible merging strategy in IterativeRS, as it significantly outperforms other methods on all metrics.
>
> |  | faithful  | summary | deberta | Avg Reward | Sharp-like |
> | :--------- | --------: | :---------: | :--------: | :------------: | :----------: |
> | RS | 0.6549 | 0.4644 | 0.3178 | 0.4791 | 3.3719 |
> | IterativeRS | 0.7023 | 0.5160 | 0.5834 | 0.6006 | 6.2180 |
>
> ## Theory
> Please note that Assumption A.3, which is the boundedness of gradients, is a standard and common assumption in convex optimization (see for example [19]). The conclusions drawn from the theoretical analysis in the paper are not guaranteed to hold for non-convex deep neural networks, which are typically used in practice. However, the goal of the theoretical analysis is not to fully explain the behavior of the proposed algorithm in practical scenarios. Instead, it demonstrates that the algorithm’s design is both sensible and grounded in a rigorous theoretical foundation. Finally, the theoretical convergence analysis of deep neural networks used in practice remains an open research problem and is beyond the scope of this paper.

---

> ### Comment · Reviewer_e3du · 2025-08-06
>
> Thanks for the response. Regarding the theory, I don't fully agree that the assumption A.3 is standard. In the reference [19] you mentioned, the assumption is applied to $\mathbb{E}[\|G\|^2]$ instead of $\|G\|^2$. For example, a Gaussian random variable $X$ can have a finite $\mathbb{E}[X^2]$ while $X^2$ is unbounded. This gap is not closed in this paper. However, I acknowledge that the paper's contribution is not theoretical. Therefore, I maintain my original score of 4 - Weak Accept.

---

> > ### Author Response · Authors · 2025-08-07
> >
> > Thank you for your feedback on our rebuttal. Regarding Assumption A3, we would like to clarify that the gradient boundedness assumption is standard and commonly used in convex optimization. If the gradient becomes unbounded, it can cause the model parameters and outputs to become unbounded as well, which in turn makes convergence impossible. Therefore, the gradient boundedness assumption is necessary for the convergence of first-order gradient descent algorithms. It is true that in many other works, the boundedness assumption is made in expectation over randomization, such as in data subset sampling. However, this still requires the gradients themselves to be bounded. Given our algorithm design, we are able to modify the assumption so that it holds in expectation, since IterativeRS is a first-order gradient descent method. Nonetheless, such modifications are not critical to the paper’s main contributions, as the theoretical analysis is provided to show that the algorithm is built on a solid theoretical foundation. Thank you again for your review and comments.

---

> > > ### Comment · Reviewer_e3du · 2025-08-07
> > >
> > > Thank you for your response. I believe you may be misunderstanding the distinction between "bounded" and "finite". Unbounded gradients can still be finite at any given point, and finite expectations do not require bounded support. A random variable can take arbitrarily large (unbounded) but finite values while maintaining a finite expectation. Therefore, your claim that boundedness in expectation requires the gradients themselves to be bounded is mathematically incorrect.
> > >
> > > However, as we both agreed, the theory is not the main contribution. Therefore, I maintain my score.

---

> > > > ### Author Response · Authors · 2025-08-08
> > > >
> > > > Thank you for your comment. In our previous comment, we meant to convey that assumption A3 is common in the sense that it is a gradient boundedness assumption. Below, we elaborate on this assumption to avoid any confusion.
> > > >
> > > > Assumption A3 states that for any data point $x$, the gradient is bounded as $\||\nabla \mathcal L(\pi_\theta(y|x))\|| \le G$. Let $\xi$ denote the subset of data points (or prompts) sampled uniformly at random at each iteration. Using assumption A3, the average gradient over the sampled subset is also bounded, with its norm less than or equal to $G$. This implies that the gradient update applied to the model parameters is bounded, contributing to the stability and convergence of the optimization process.
> > > >
> > > > In many studies the assumption is on the expectation meaning that the boundedness assumption is as $ \mathbb E[\||\nabla \mathcal L(\pi_\theta(y|x))\||] \le G$ where the expectation is taken with respect to randomization in data sampling. This assumption similarly ensures that the expected gradient update is bounded.
> > > >
> > > > Adapting Assumption A3 to the expectation form is straightforward, given that IterativeRS is a first-order gradient descent algorithm. However, for simplicity and clarity of notation, we use the current form. Since the choice of data subset does not play a critical role in the proposed algorithm, prior analyses on the influence of data subsampling in gradient-based optimization apply directly to our setting. To clarify this in the paper, we can add a brief comment.

---

### Decision · Program_Chairs · 2025-09-17

**Decision:**

Accept (poster)

**Comment:**

This paper proposes a method for RL fine-tuning of foundation models using several rewards, which the authors term IterativeRS. This is achieved by carefully controlling the trade-off between specializing policies towards different rewards and fusing them into a single shared policy.



On the positive side, all reviewers generally considered the paper technically sound and well-written. They noted that IterativeRS elegantly unifies two existing RL approaches (RS on one end, and MORLHF on the other). Most reviewers also praised the diversity of domains in which IterativeRS is evaluated.



On the other hand, some reviewers were unconvinced by relatively modest performance improvements in some settings; although during rebuttal authors provided improved results by tuning the task weighing, this approach only yielded improvements in a subset of domains. Some reviewers also considered the novelty of the work to be somewhat limited.



After rebuttal, four reviewers were leaning towards acceptance, while one (Reviewer ysgq) towards rejection. In subsequent Reviewer-AC discussion, the negative reviewer kept their score, but also was not strongly opposed to acceptance.



Taking all of the above into account, I believe the paper is technically sound, has extensive evaluation, offers modest but consistent performance gains, and is somewhat novel although heavily building on prior work. I recommend acceptance, and encourage authors to take reviewer feedback into account when preparing a camera-ready version of the paper.